



# A quest for precipitation attractors
# in weather radar archives

Loris Foresti[1], Bernat Puigdomènech Treserras[2], Daniele Nerini[1], Aitor Atencia[3],
Marco Gabella[1], Ioannis V. Sideris[1], Urs Germann[1], and Isztar Zawadzki[2†]

[1]Federal Office of Meteorology and Climatology MeteoSwiss, Locarno-Monti, Switzerland
[2]Department of Atmospheric and Oceanic Sciences, McGill University, Montreal, Canada
[3]GeoSphere Austria, Vienna, Austria

**Correspondence:** Loris Foresti (loris.foresti@meteoswiss.ch)

**Abstract.** Archives of composite weather radar images represent an invaluable resource to study the predictability of precipitation. In this paper, we compare two distinct approaches to construct empirical low-dimensional attractors from radar precipitation fields. In the first approach, the phase space dimensions of the attractor are defined using the domain-scale statistics of precipitation fields, such as the mean precipitation, fraction of rain, spatial and temporal correlations. The second type of attractor considers the spatial distribution of precipitation and is built by principal component analysis (PCA). For both attractors, we investigate the density of trajectories in phase space, growth of errors from analogue states, and fractal properties. To represent different scales, climatic and orographic conditions, the analyses are done using multi-year radar archives over the continental United States ($\approx 4000 \times 4000$ km$^2$, 21 years) and the Swiss Alpine region ($\approx 500 \times 500$ km$^2$, 6 years).

## 1 Introduction

Precipitation is challenging to forecast. The difficulty is due to its large space-time variability (e.g. Lovejoy and Schertzer, 2013), the many non-linear processes involved (e.g. Houze, 2014) and the resulting chaotic behaviour of the atmosphere (e.g. Lorenz, 1963), among others.

As a result, a rapid loss of precipitation predictability is observed for both extrapolation-based nowcasting and NWP-based forecasting (e.g. Surcel et al., 2015). Such limits to predictability drive the need for more accurate estimates of forecast uncertainty to enable informed decision making.

Lorenz (1996) defines two types of *predictability*:

- *Intrinsic predictability*: "the extent to which prediction is possible if an optimum procedure is used".

- *Practical predictability*: "the extent to which we are able to predict by the best-known procedures".

The goal of forecasting is to design models whose practical predictability is as close as possible to the intrinsic predictability while representing the remaining uncertainty.

---

† Passed away on 11 February 2023.





Studies on atmospheric predictability are either model-based or observation-based; see a review in Lorenz (1996) and Germann et al. (2006b). Modeling studies use either idealized systems of equations (e.g. Lorenz, 1963) or NWP models (e.g. Palmer and Hagedorn, 2006). One disadvantage of such methods is related to the strong assumptions on how precipitation processes are represented in the models.

Observation-based predictability studies comprise statistical extrapolation methods (e.g. Germann et al., 2006b) and naturally occurring analogues (e.g. Lorenz, 1969). Common challenges are related to the presence of measurement uncertainty, the assumption of attractor smoothness (e.g. Takens, 1981), and the limited size of archives (e.g. Toth, 1991; Van Den Dool, 1994), which only allows finding analogues of 'mediocre' quality (Lorenz, 1969). Precipitation brings further challenges due to its truncated non-Gaussian distribution, intermittent and multifractal properties (Schertzer and Lovejoy, 1987; Lovejoy and Schertzer, 2013).

Atencia and Zawadzki (2017) used the Lorenz63 system to compare the growth of errors (spread) from analogue states with the one obtained from standard perturbation techniques used in NWP ensemble forecasting. They showed that analogues display a similar initial error growth, but contain more information throughout the forecast. Therefore, despite the limitations of observation-based studies, analogues have good potential to complement predictability studies.

Inspired by Atencia et al. (2013), this study aims to construct low-dimensional attractors from weather radar archives to shed new light on the intrinsic predictability of precipitation. We compare two distinct approaches to construct the attractor. The first is a *deductive* approach based on prior knowledge, that is, the phase space dimensions are defined based on domain expertise and (forecast) application requirements. The second is an *inductive* approach, where the phase space dimensions are extracted from the data without prior assumptions (except those required by PCA). This research has an exploratory character and focuses on what worked and did not work in our quest for the precipitation attractor. The careful reader will notice that the methodology of the Swiss and US attractors is not 100% consistent. Indeed, most of the research started independently and only converged into this paper at a later stage. The investigation of the differences triggered fruitful discussions.

In this paper, we want to answer the following questions:

- What do we learn about the predictability of precipitation from weather radar archives?

- How to define the phase space of the attractor?

- What is the typical growth of errors from analogues?

- How does predictability depend on scale?

- In which way is the attractor relevant for short-term precipitation forecasting and stochastic simulation?

The paper is structured as follows. Section 2 describes the two radar archives and the conceptual framework. Section 3 defines the attractor based on domain-scale statistics and presents analyses of its properties. The same is done in Sect. 4 using principal component analysis. Section 5 concludes the paper and discusses future perspectives. Statistical techniques are detailed in Appendix.




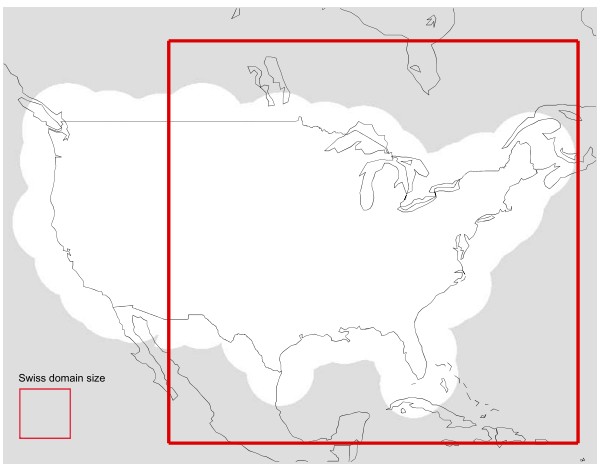

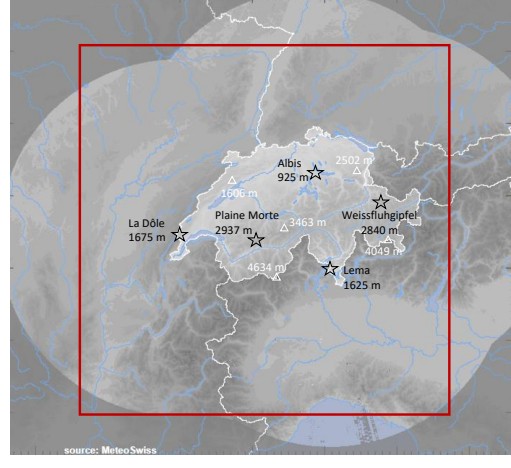

**Figure 1.** a) Continental US analysis domain of $4096 \times 4096$ km$^2$ (white background indicates the radar composite coverage). b) Swiss analysis domain of $512 \times 512$ km$^2$ (lighter gray background indicates the radar composite coverage, the black text the location and height of weather radars, and the white text the location and height of important mountain peaks). The US domain surface is 64 times larger than the Swiss domain.

**Table 1.** Characteristics of the Swiss and US radar datasets. *: number of images with wet area ratio $\geq 5$ %.

| Domain | United States | Switzerland |
|---|---|---|
| Domain size | $4096 \times 4096$ km$^2$ | $512 \times 512$ km$^2$ |
| Grid points (M) | 1'048'576 | 262'144 |
| Spatial resolution | 4 km | 1 km |
| Temporal resolution | 15 min | 5 min |
| Period | 1996-2016 | 2005-2010 |
| N images (with precip.) | $\approx 700'000$ | $\approx 210'000$* |
| N images (theor. max) | 736'416 | 631'008 |

## 2 Data and conceptual framework

### 2.1 Archives of composite radar images

The US and Swiss radar datasets are described in Table 1. The US data are produced by the operational S-band Weather Surveillance Radar-1988 Doppler network (WSR-88D) covering the continental United States (CONUS). The archive spans a 21-year period from 1996 to 2016 and was obtained by interpolating four different radar composite products to a common 4 km resolution grid. Radar products comprise the maximum echo from any radar (1996-2007) as well as more advanced products removing ground clutter and blending of multiple radars, see more details in Atencia and Zawadzki (2015) and Fabry

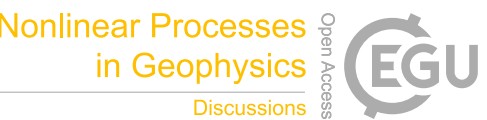

et al. (2017). For computational reasons, the temporal resolution was reduced from 5 min to 15 min and a smaller domain of $4096 \times 4096$ km$^2$ was extracted (see Fig. 1). When needed, data were upscaled by averaging rainrate in linear units using the Marshall-Palmer Z-R relationship $Z = 300R^{1.5}$. Note that radar visibility on the Rocky mountains is rather limited by the large inter-radar distances, which is the reason why the domain is cut (see Appendix E).

The Swiss data comprise the Quantitative Precipitation Estimation (QPE) product, which integrates measurements from three Doppler C-band weather radars (Germann et al., 2006a). The archive covers a 6-year period from 2005 to 2010 and has a spatial resolution of 1 km and a temporal resolution of 5 min. The domain was reduced to a square 512 km grid centred over Switzerland (see Fig. 3a). The radar network was upgraded to dual-polarization in 2011 and equipped with two new radars in 2014 and 2016 to improve the coverage in the inner Alpine valleys (e.g. Germann et al., 2022). To avoid temporal

inhomogeneity in the archive introduced by the switch to the new radar generation, in this study we did not include data after 2011.

Radar-based quantitative precipitation estimation (QPE) is inevitably affected by uncertainty due to, e.g. the Z-R relationship, variability of the vertical profile of reflectivity, signal attenuation, and residual clutter (e.g. Villarini and Krajewski, 2010). However, these uncertainties are not expected to substantially alter the main findings of this paper, as we concluded in a related

paper using Swiss radar data (Foresti et al., 2018).

## 2.2    Is the radar archive large enough?

Inspired by Van Den Dool (1994), in Fig. 2 we estimated the minimum size of the radar archive needed to obtain sufficiently good analogues using the US dataset by targeting a spatial resolution of 4 km. The methodology consists of:

1. calculating the correlation dimension log-log plot for increasing archive sizes (see Appendix C and D),

2. detecting the 'crossing' points, i.e. the smallest scaling distance between real analogues for increasing archive sizes (squares in Fig. 2a),

3. selecting the largest correlation dimension and extrapolating the linear fit to obtain the correlation integral $C_r$ associated to the desired resolution of $r = 4km$, i.e. $C_{4km} = 10^{-30}$ ($\approx$ radar observation error), (Fig. 2b),

4. plotting the archive size vs the $C_r$ values of the crossing points and extrapolating the resulting fit to the desired $C_{4km}$ to

obtain the required archive size (Fig. 2c).

The point of crossing is detected on the $log(r) - log(C_r)$ curve (Fig. 2a). It represents the point of maximum curvature between the steep scaling region in the middle of the curve and the flat region on its left (small radii). The flattening at small scales is due to the temporal correlation of the data, i.e. the consequence of estimating $C_r$ on temporally correlated points (trajectories) rather than independent points. In contrast, the flattening on the right of the curve (large radii) occurs when the radii become

larger than the subspace occupied by the attractor.

For $C_{4km} = 10^{-30}$ the number of points required to find good analogues is $3.41 \cdot 10^{26}$, which corresponds to $9.73 \cdot 10^{21}$ years! This result is based on the degrees of freedom of the image, which corresponds to the number of pixels. Consequently, the only



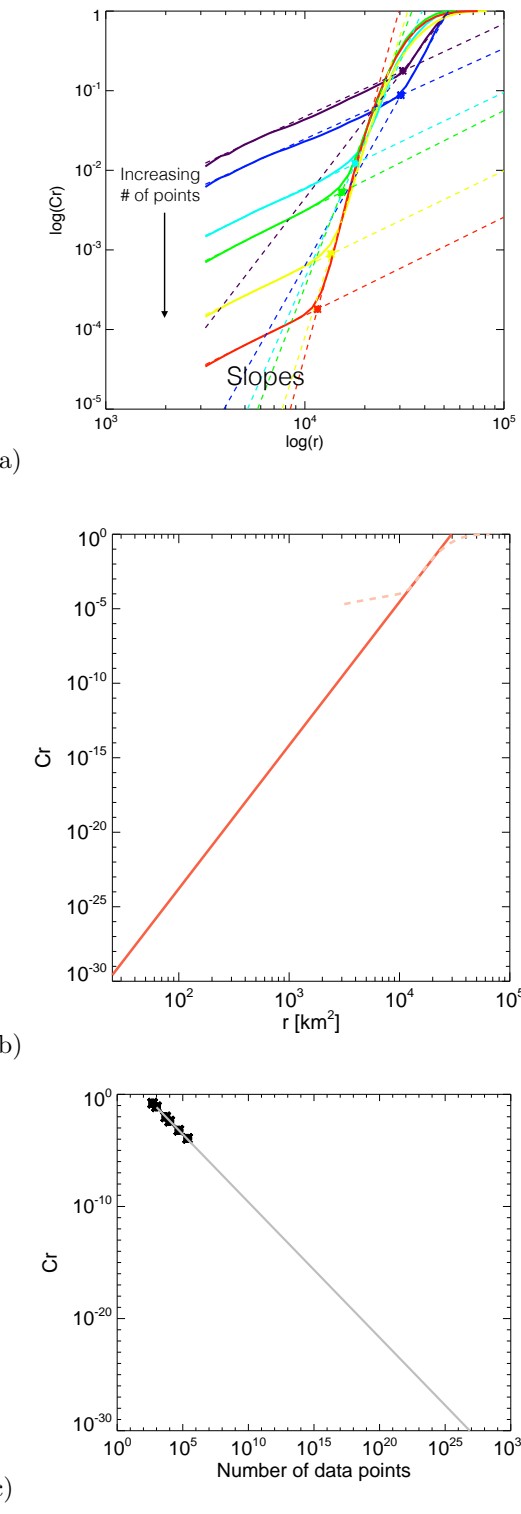

**Figure 2.** Estimation of the theoretical size of archive needed to find good radar analogues at 4 km resolution in the US. a) Estimation of correlation dimension and the point of crossing for increasing archive sizes (purple to red colors). b) Estimation of the correlation dimension associated to a distance $r = 4$. c) Estimation of the archive size needed to reach $C_{4km}$ from the points of crossing.





way to find "good" analogues from the radar datasets is to reduce the degrees of freedom by defining a lower-dimensional phase space, which is the main objective of this paper.

### 2.3 From a high-dimensional dataset to a low-dimensional attractor

An archive of radar rainfall fields can be structured as a temporal sequence of images into a 2D array of size $N \times M$:

$$\mathbf{X}_{N,M} = \begin{pmatrix} x_{1,1} & x_{1,2} & \cdots & x_{1,M} \\ x_{2,1} & x_{2,2} & \cdots & x_{2,M} \\ \vdots & \vdots & \ddots & \vdots \\ x_{N,1} & x_{N,2} & \cdots & x_{N,M} \end{pmatrix} \tag{1}$$

where $x_{1,2}$ is the rainrate at time index 1 and pixel index 2, $N$ is the number radar rainfall fields, and $M$ is the number of grid points within a field, i.e. each row is a flattened radar image. Hence, a sequence of rainfall fields represents a trajectory in an M-dimensional phase space, where $M = 512 \times 512 = 262144$ for the Swiss domain and $M = 1024 \times 1024 = 1048576$ for the US domain.

A common approach to study nonlinear dynamical systems is to look at the evolution of trajectories in the phase space of governing variables (e.g. Lorenz, 1963; Abarbanel, 1997; Kantz and Schreiber, 2004). In this paper, the governing variables of the precipitation system are assumed to be the domain-scale statistics or principal components extracted from the high-dimensional radar archive.

The attractor represents the subspace attracting the trajectories of atmospheric states, in our case radar precipitation fields, starting from any initial condition within the phase space. Chaotic systems, i.e. systems with sensitive dependence on initial conditions, never cross the same trajectory again and generate "strange" attractors, which have a non-integer intrinsic dimension (fractal dimension). The Lorenz system and the atmosphere are two examples of strange attractors. The strange attractor of this study is the ensemble of possible states and trajectories derived from weather radar images that are consistent with the precipitation climatology of a given region.

### 2.4 Measuring error growth

As time passes divergence among two initially close states in phase space increases, which is generally referred to as *error growth* (or spread growth). Idealized chaotic systems show three distinct regions of error growth: 1) an exponential growth at the start, 2) a power-law growth, and 3) a region of saturation (loss of predictability); see for example Nicolis et al. (1995); Atencia and Zawadzki (2017).

The established way to estimate the initial growth of errors, and therefore inferring the predictability of the system, is to compute Lyapunov exponents (e.g. Abarbanel, 1997; Kantz and Schreiber, 2004). Lyapunov exponents measure the rate of exponential error growth assuming an infinitesimally small initial error and an infinite lead time, while the maximum Lyapunov exponent is a measure of chaos strength (e.g. Lichtenberg and Lieberman, 1992). Unfortunately, such conditions are only met in idealized systems of equations (e.g. De Cruz et al., 2018). Such conditions are never observed with atmospheric





measurements, whose analogue states have too high initial errors and measurement noise. Therefore, in this paper we looked for practical alternatives to Lyapunov exponents. Depending on the experiment we computed as a function of lead time the:

- *Standard deviation ($\sigma$) of analogues* around their mean:

$$s_1(t) = \sqrt{\frac{1}{N-1} \sum_{i=1}^{N} ||\mathbf{x}_i(t) - \overline{\mathbf{x}(t)}||^2}. \tag{2}$$

- *Half the difference between the $84^{th}$ and $16^{th}$ quantiles* of the distribution of analogue states:

$$s_2(t) = [\mathcal{Q}_{84}(t) - \mathcal{Q}_{16}(t)]/2. \tag{3}$$

which is analogous to $s_1$ for a Gaussian distribution.

The analyses on the US attractor used Eq. 2, while the ones on the Swiss attractor used Eq. 3. For simplicity, we refer to them as *spread* or *error*. The spread was further normalized by the sample climatological spread computed over the whole archive so that saturation is reached around the value of 1 (comparable to selecting random analogues). The techniques to retrieve two (or more) close states are briefly described in the corresponding Sections presenting the results.

### 2.5 Fractal properties of the attractor

Fractal properties of the attractor can give useful insights into its intrinsic dimensionality. In this paper, we combined the time-delay embedding and Grassberger-Procaccia correlation dimension methods (Grassberger and Procaccia, 1983).

The approach works by iteratively increasing the dimensionality of embedding space $D$ (by time-delay) until the estimated fractal dimension $f$ (by Grassberger-Procaccia) converges to a finite value. When this point is reached, the attractor is completely unfolded, which *may* be evidence of a system driven by low-dimensional chaotic dynamics (and thus characterized by predictability). In contrast, the inability to converge towards a finite dimension would indicate the presence of unpredictable stochastic processes. Appendix C and D describe the time-delay and Grassberger-Procaccia methods and provide some references describing their known limitations.

## 3 Precipitation attractors based on domain-scale statistics

### 3.1 Extracting phase space dimensions

There are many summary spatial and temporal statistics that can be extracted from precipitation fields and used as phase space dimensions, which we refer here to as *domain-scale statistics*. Keeping in mind that the attractor will be used for precipitation nowcasting/forecasting, we aim to extract dimensions that are relevant for those specific tasks. Several probabilistic precipitation nowcasting systems generate an ensemble nowcast by adding spatially- and temporally-correlated random perturbations to a deterministic extrapolation (e.g. Pegram and Clothier, 2001; Bowler et al., 2006; Berenguer et al., 2011; Atencia and





Zawadzki, 2014; Nerini et al., 2017; Pulkkinen et al., 2019; Sideris et al., 2020). Such stochastic methods typically need the
150 reproduce the following properties of precipitation fields:

- the *Fourier transform* of the field (power spectrum), which is used to generate stochastic precipitation fields with a given
  spatial auto-correlation (e.g. Schertzer and Lovejoy, 1987; Pegram and Clothier, 2001),

- the *fraction of precipitation*, which imposes the correct amount of zero precipitation (intermittency) on the spatially-
  correlated field,

- the *mean precipitation*, which re-scale the non-zero values to reproduce the observed precipitation distribution,

- the *temporal auto-correlation* of precipitation fields, which is used by auto-regressive processes to make precipitation
  fields evolve over time.

A radially-averaged 1D power spectrum (RAPS) can be derived from the 2D Fourier power spectrum (see Appendix A). The
RAPS is one simple approach to check if the precipitation field exhibits scale invariance within a given range of spatial scales,
which is manifested in a power-law relationship between the logarithm of the scale (spatial frequency) and the logarithm of the
power spectrum:

$$P(k) \propto k^{-\beta},$$
$$\log\big(P(k)\big) \propto \beta \log\big(k\big), \tag{4}$$

where $P$ is the Fourier power spectral density, $k$ the spatial frequency, and $\beta$ is the slope of the power law, called *scaling expo-*
165 *nent* or *spectral slope*. $\beta$ can be derived by ordinary least squares from the log-log plot of frequency against power. The Fourier
transform can only account for the scaling of the second moment (variance), known as simple scaling. Multifractal approaches
can handle the scaling of higher-order moments together with the intermittency of the field within a unified framework (e.g.
Lovejoy and Schertzer, 2013).

Because rainfall rates often follow a log-normal distribution, it is more convenient to perform the Fourier transform on the re-
170 flectivity ($Z$) or rainfall rate ($R$) transformed in multiplicative units, i.e. $dBZ = 10\log_{10}(Z/Z_0)$ and $dBR = 10\log_{10}(R/R_0)$,
respectively, where $Z_0 = 1mm^6/m^3$ and $R_0 = 1mm/h$. In addition, before performing the Fourier transform we advise to set
all values below a chosen minimum threshold to the minimum threshold itself (e.g. $x_i < 0.1 \mapsto 0.1\,mm^{-1}, \forall i$). This operation
removes weak precipitation signals and smooths the resulting sharp corners of the rain/no-rain transition, which reduces the
overestimation of power at high frequencies $\beta$ (e.g. Nerini et al., 2017).
Several summary, spatial and temporal statistics were derived for the Swiss and US attractors, for instance:

- WAR: wet area ratio (Pegram and Clothier, 2001). Percentage of wet (rainy) pixels over the radar composite domain ($\geq$
  0.1 mm h$^{-1}$).

- Area coverage: number of wet pixels over the radar composite domain. This is similar to WAR.



- IMF: image mean flux (Pegram and Clothier, 2001). Unconditional mean precipitation (including zeros). Note that this variable is correlated to WAR.

- MM: marginal mean precipitation. Conditional mean precipitation (only wet pixels). Also referred to as conditional IMF. It can be computed in rainrate (mm h$^{-1}$) or in linear reflectivity units $Z$.

- $\beta_1$ and $\beta_2$: slopes of the spatial 1D RAPS. Two values are needed since spectra often show a scale break (e.g. Gires et al., 2011; Seed et al., 2013).

- $e$: Anisotropy of the precipitation field, as measured by the eccentricity of the spatial autocorrelation function. The latter is derived as $e = \sqrt{1 - a_m/a_M}$, where $a_m$ and $a_M$ are the minor and major axes of the fitted ellipse (derived by eigenvalue decomposition of the spatial covariance matrix).

- Decorrelation time of precipitation fields, defined as the time when the temporal correlation falls below the value $1/e \approx 0.37$.

Figure 3 shows an example of a composite radar rainfall field at 1700 UTC on 17 April 2016, Switzerland, with the corresponding 2D power spectrum, 1D RAPS, and 2D autocorrelation function (computed by inverse FFT of the 2D spectrum, see Appendix A). The RAPS exhibits the typical power law scaling behavior of precipitation fields. The scaling break occurs around the 20 km wavelength, in agreement with other studies (Seed, 2003; Seed et al., 2013).

## 3.2 Phase space trajectories

Figure 4 represents the density of points (trajectories) of a 4D US precipitation attractor that was constructed using the following phase space dimensions: decorrelation time, eccentricity, area coverage and marginal mean. These dimensions were selected to be independent from each other, although we can still notice some interesting dependencies between variables, for example the increase of decorrelation time with increasing rainfall area (top-left) and increasing eccentricity (top-center). This may indicate that precipitation fields that are less widespread and more isotropic, for example isolated convective cells, are less predictable (according to the decorrelation time), while more organized frontal precipitation with large scale anisotropy is more predictable. The density plot of decorrelation time versus marginal mean also has a peculiar shape. The decorrelation time increases until MM $\approx 23$ dBZ, but then starts decreasing. This could be attributed again to the lower predictability of isolated intense convective cells compared to stratiform precipitation. Moreover, the larger variance of convective precipitation reduces the decorrelation time.

Figure 5 visualizes a Swiss 4D precipitation attractor embedded in the phase space composed of WAR, MM, $\beta_1$ and $\beta_2$. These variables are different from the ones used in the US because in Switzerland we were developing a nowcasting system depending on those 4 variables, while in Canada we were doing a more general purpose study. The attractor is constructed using all the rainfall fields from 2005 to 2010 that have a WAR $\geq 5$ % and where the fitting of the spectral slopes is of good quality, i.e. when the correlation coefficient of the linear regression is above 0.95 (for both $\beta_1$ and $\beta_2$). These criteria are met



**Figure 3.** Fourier analysis of the radar rainfall field at 1700 UTC on 16 April 2016, Switzerland. a) Radar rainfall field overlaid on the Digital Elevation Model (DEM); b) 2D Fourier power spectrum rotated by 90° (zoom for wavelengths larger than 13 km); c) Spatial autocorrelation function; d) Radially averaged 1D power spectrum in a log-log plot, together with the estimated spectral slopes $\beta_1$, $\beta_2$, WAR, IMF and MM statistics. See other examples in Nerini et al. (2017).

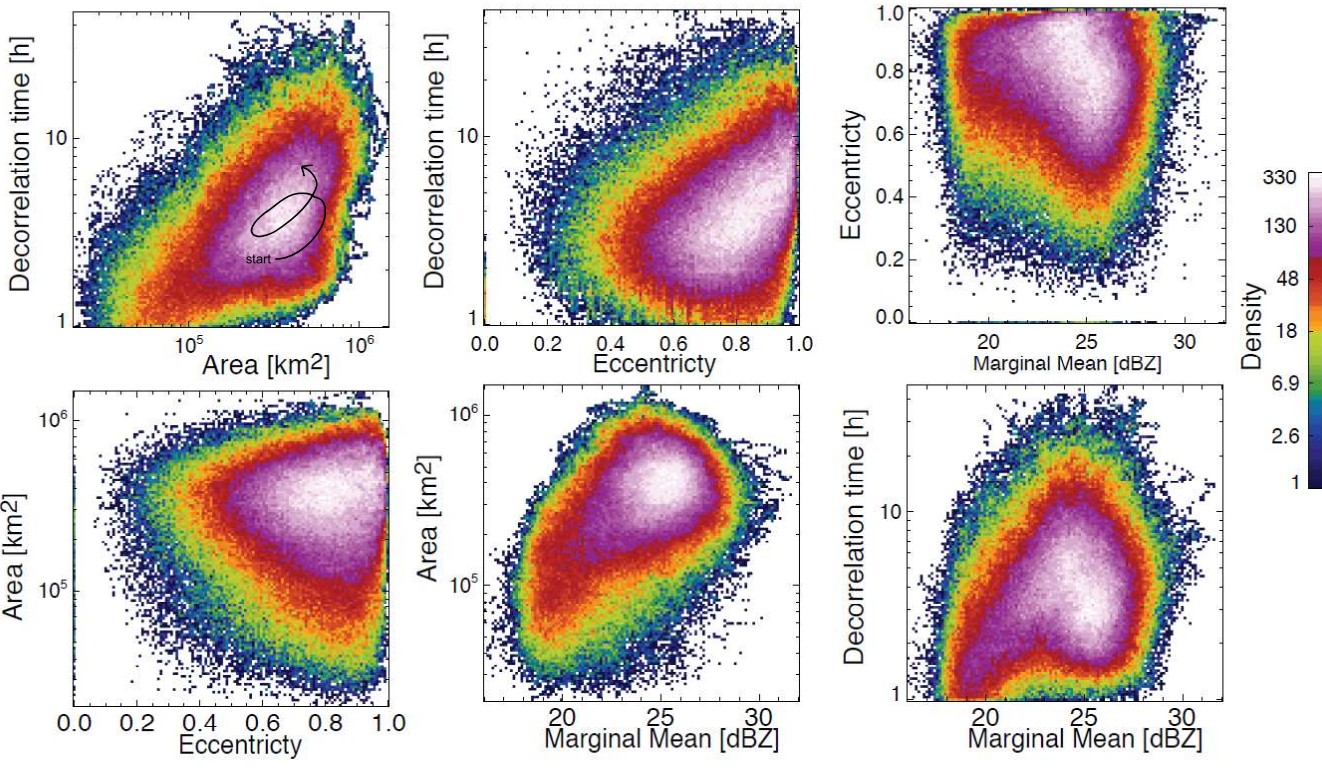

**Figure 4.** US precipitation attractor. It is represented by 2D histograms (counts) using as phase space dimensions the marginal mean precipitation (MM), area coverage, decorrelation time and eccentricity of composite radar precipitation fields. For illustrative purposes, the upper left panel includes a dummy trajectory representing a sequence of radar fields within the attractor.

by 209'715 rainfall fields. The figure panels are organized in a $4 \times 4$ matrix, where each row (column) represents one phase space dimension.

The 4 plots on the diagonal show the univariate histograms of the 4 variables and the corresponding summary statistics (mean and standard deviation).

The subplots on the upper triangular part of the matrix show the 2D histograms describing the density of points for all combinations of phase space variables (same as Fig. 4). In the upper right part of each subplot, there is the correlation coefficient between the two variables. Interesting correlations can be noticed between WAR vs $\beta_1$ and MM vs $\beta_2$. The first reveals that increasing the rainfall fraction over the radar domain increases the power at large spatial scales. The second highlights the







**Figure 5.** Swiss 4D precipitation attractor. The phase space dimensions are the marginal mean precipitation, the wet area ratio, and the two spectral slopes of the RAPS ($\beta_1$, $\beta_2$). MM and WAR are shown in log scale to account for the asymmetry of the distribution.




convective cases (high MM), which increase the power at wavelengths of $\approx$ 10-20 km and thus the value of $\beta_2$. In the context of spatial scaling analysis, the density plot of $\beta_1$ vs $\beta_2$ is quite interesting as it summarizes the average scaling behavior of

Alpine radar rainfall fields over many years, i.e. $1.8 < \beta_1 < 2.4$ and $3.3 < \beta_2 < 3.8$. Such findings are relevant in the context of stochastic rainfall simulation since $\beta$ determines the type of approach needed, which depends on whether $\beta$ is below or above the dimension of the field ($\beta > 2$), see e.g. Menabde (1998).

    The subplots in the lower triangular part of the matrix are a simplified representation of *surfaces of section*, also known as Poincaré maps (e.g. Lichtenberg and Lieberman, 1992). They are obtained by plotting all the points that intersect a given

surface of section (e.g. Sideris, 2006), which is defined here by a small interval around the 50-percentile of a given variable, e.g. in the range 2.13-2.16 for $\beta_1$ (for MM vs WAR). The points selected within that interval are colored according to the remaining $4^{th}$ variable (e.g. $\beta_2$), which helps analyzing the variable dependencies in the 4D space, e.g. the clear increase in $\beta_2$ when increasing MM (row 3, col 2) or the increase in $\beta_2$ with decreasing $\beta_1$ (row 3, col 1), which was not visible from the density plot (row 3, col 4).

These graphical illustrations represent a first useful insight into the attractor. For example, it is possible to distinguish the statiform and convective precipitation systems using combinations of phase space dimensions, in particular the eccentricity, spectral slopes and decorrelation time.

### 3.3 Scaling properties

The domain-scale precipitation attractor provides additional insight into the origin of the scaling break in the Fourier power

spectrum, which was already noticed by previous studies, e.g. Gires et al. (2011) and Seed et al. (2013).

    Figure 6 shows the relationship between the magnitude of the scaling break, defined as $\beta_2 - \beta_1$, with the rainfall fraction (WAR). Figure 6a shows that the scaling break magnitude tends to decrease for increasing WAR values. This dependence is enhanced further when normalizing the WAR by the MM, which leads to a correlation of almost -0.5 (Fig. 6b). In other words, the scaling break is more pronounced when intense precipitation is concentrated in a few areas (small WAR and high MM). In

contrast, widespread low-intensity precipitation reduces the magnitude of the scaling break.

    This brief analysis sheds new light in the origin of the scaling break in power spectra of precipitation fields, which is helpful to design stochastic models (e.g. Seed et al., 2013). For instance, one single spectral slope would be sufficient to simulate stratiform precipitation fields, while two spectral slopes are necessary to simulate convective precipitation fields.

### 3.4 Fractal properties

To estimate the fractal dimension of the attractor, we applied the time-delay embedding technique and correlation dimension method to each time series of phase space dimensions.

    Figure 7 shows the estimated fractal dimension of the US precipitation attractor for the fractional area coverage and the marginal mean time series. For an embedding space of $D = 30$ dimensions, the correlation dimensions stabilize to $f$=9.85 for the fractional area and $f$=10.89 for the marginal mean. However, due to known limitations of the Grassberger-Procaccia



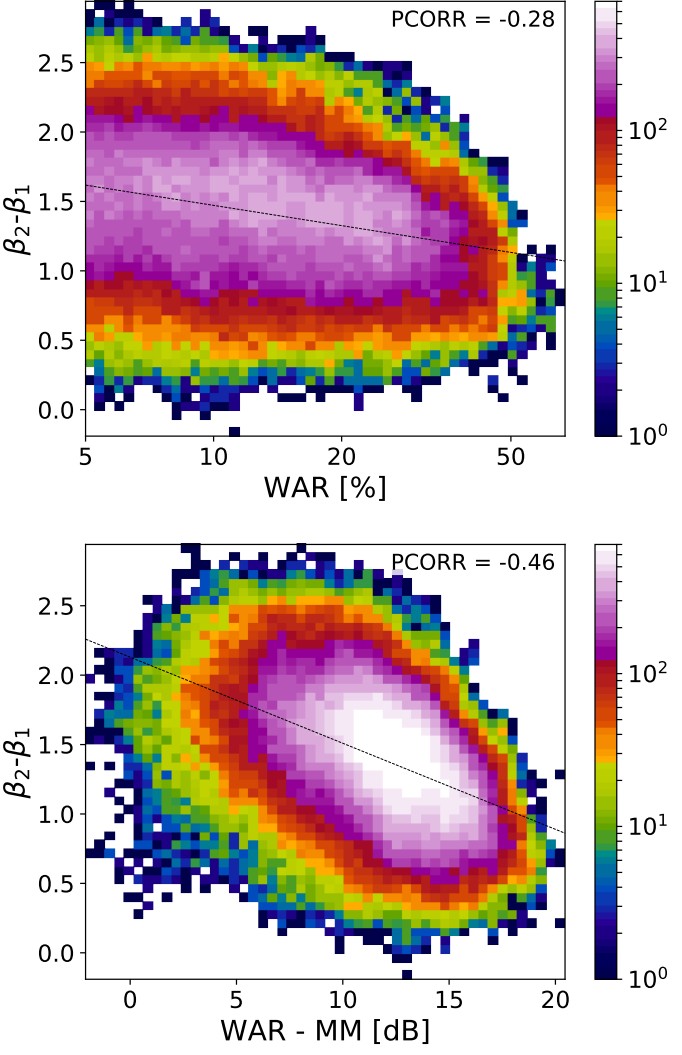

**Figure 6.** Analysis of scaling break magnitude ($\beta_2 - \beta_1$) w.r.t. (normalized) rainfall fraction. 2D histogram of a) $\beta_2 - \beta_1$ vs WAR [%, log scale] and b) $\beta_2 - \beta_1$ vs WAR-MM [dB]. If $\beta_2 - \beta_1 = 0$ there is no scaling break.



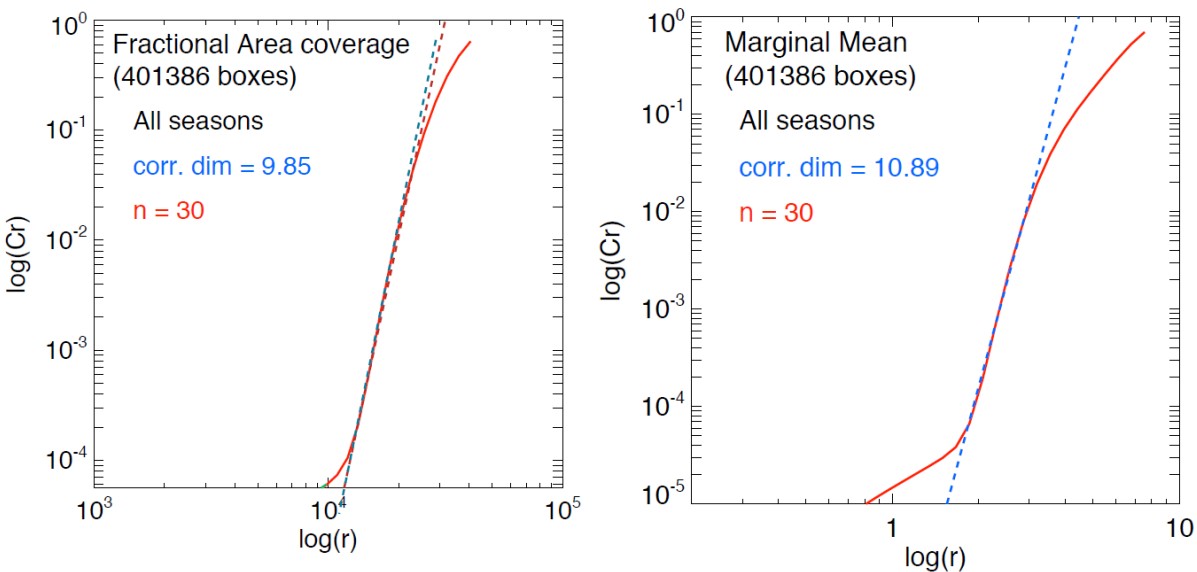

**Figure 7.** Correlation dimension estimation using the Grassberger-Procaccia algorithm and time-delay embedding on the variables fractional area coverage and marginal mean on the US attractor. $log(r)$ is the logarithm of the radius containing the points, while $log(Cr)$ is the corresponding (normalized) correlation integral, i.e. the average number of points found within that radius.

algorithm, we cannot claim that such finite correlation dimensions are the evidence of a low-dimensional chaotic system (see references in Appendix D).

A similar analysis was also done for the Swiss attractor but within the PCA framework, which proved more useful (see Section 4.6).

### 3.5 Growth of errors

Once the phase space is defined, we can study the intrinsic predictability of states starting from close initial conditions, the so called *analogues*.

Figure 8 shows the average growth of the standard deviation of analogues (spread) on the US attractor. The analyses are done by incrementally adding phase space dimensions starting from the rainfall area. The error growth (spread) is characterized by the following stages:

1. $\approx$0-1h: initial slow exponential growth (nowcasting range),

2. $\approx$1-6h: fast power-law growth (from nowcasting to short range),

3. $\approx$6h-20d: slow growth again (from short range to medium range),

4. >$\approx$20d: saturation stage (loss of predictability).

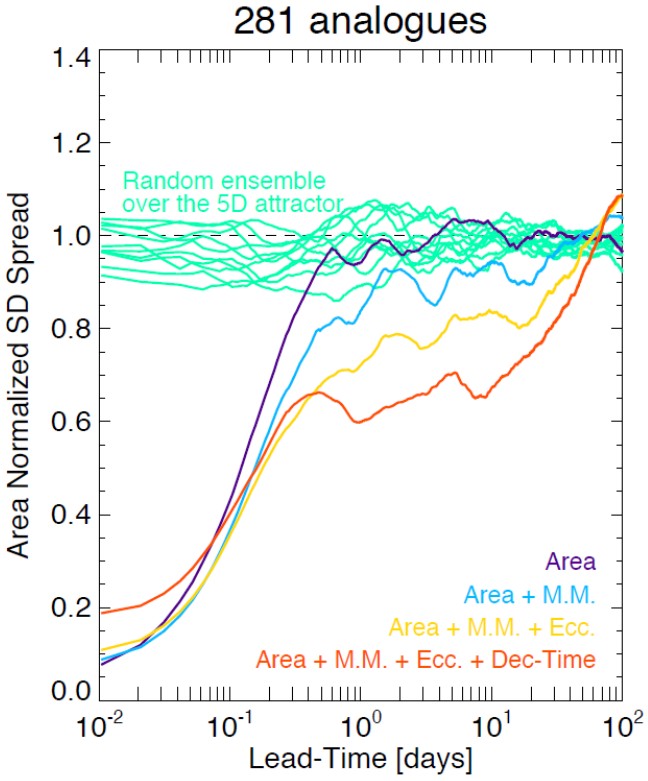

**Figure 8.** Average standard deviation of analogues in the US attractor as a function of lead time for phase spaces of increasing dimensionality. The x-axis is logarithmic while the y-axis is linear.

The reason for slow error growth at 0-1h is mostly unknown. It might be related to some radar data processing steps, in particular those that introduce smoothness in the precipitation field, which leads to overestimating the predictability at the smallest scales. The rapid error growth at 1-6h is attributed to the low predictability of precipitation growth and decay, especially in convective systems. The slower error growth at 6h-20d can be explained by the more predictable translation of synoptic scale features across continental US.

Note that saturation already occurs after ≈6h when using the variable Area alone. Adding phase space dimensions improves predictability in three ways: 1) by reducing the rate of error growth in the first 2-3 hours, 2) by reducing the spread in range ≈6h-20 days, and 3) by extending the saturation stage by several days. However, the 4D attractor has a larger initial error ($\approx 0.2$), which reflects the difficulty of finding states that are similar in all dimensions simultaneously.

These promising results show that there is unexpected intrinsic predictability of precipitation if the appropriate phase space dimensions are chosen. Given the substantial improvement of predictability in range ≈6h-20 days, it would also be interesting to study whether analogues can help extending the range of NWP models (e.g. by blending probabilities).





**Figure 9.** Error growth of an ensemble of analogues starting from different initial conditions on the Swiss attractor. The upper end of the x-axis corresponds to a lead time of 96 hours (4 days). The number of analogues $N$ is shown in the figure legends. Plots are in log-log scale.





Another interesting experiment is to analyze the local variability of predictability within the attractor. This could inform about the dependence of intrinsic predictability on initial location within the attractor (e.g. Li and Ding, 2011).

In order to simplify the task, we consider here only 1D trajectories, i.e. time series of individual phase space variables, hereafter extracted from the Swiss attractor. The small interval defining the initial conditions is selected by regularly spaced values between the $20^{th}$ and $90^{th}$ quantiles of each phase space variable. At each quantile, we select all the analogues that are within a small neighborhood and that are at least 1h from each other (to reduce dependence among analogues).

Figure 9 shows the growth of spread of an ensemble of analogue time series starting at close initial conditions. Instead of using a log-linear scale as in Fig. 8, here we use a log-log scale to highlight the power-law growth of errors. The reported lifetime characterizes the lead time after which the spread reaches saturation. It is estimated by fitting a non-parametric kernel ridge regression to the data points and by taking the value at which the first derivative approaches zero. Such estimations are only approximate as they depend on the convergence criterion chosen.

Depending on the phase space variable and the initial conditions we obtain different lifetime estimations, which reflects the predictability dependence on initial conditions. The saturation times vary between 2h and more than 10h, which are substantially shorter than those obtained in the US (Fig. 8). The shorter predictability over the Alpine region is mainly attributed to the smaller domain size, which is 64 times smaller than the US.

Most curves in the Swiss attractor miss the initial slow error growth observed in the US attractor (Fig. 8). They also start from quite different initial values depending on the variable chosen. For example, $\beta_2$ has rather high initial errors, which indicates that it has a larger intrinsic variability (noise). Despite the larger initial error, $\beta_2$ seems to take longer than $\beta_1$ to saturate (from several hours to days depending on the chosen saturation thresholds). Despite flattening considerably after 5-6 hours, the MM does not reach saturation completely.

These findings provide some useful information on the predictability that could be obtained by extrapolation nowcasts. In fact, the smaller size of the Swiss domain (as compared with the US) imposes a shorter limit of predictability, which could only be extended by enlarging the domain or by forecasting the evolution of domain-scale statistics.

## 4 Precipitation attractors based on principal component analysis

### 4.1 Extracting phase space dimensions

Domain-scale statistics are unable to describe the spatial distribution of precipitation, unless they are computed locally (e.g. Sideris et al., 2020). A possible solution is to use Principal Component Analysis (PCA). PCA is a method to compress the information contained in a dataset of correlated variables, which has been extensively used in atmospheric and climate science (e.g. Lorenz, 1956; Richman, 1986; Jolliffe, 2002; Schiemann et al., 2010; Foresti et al., 2015; Nerini et al., 2019). The procedure consists of finding an orthogonal transformation that linearly combines the variables to form a set of uncorrelated variables sorted by explained variance, which are called principal components.

The variables of the precipitation data matrix (Eq. 1) are strongly correlated since each column represents a time series of precipitation for one pixel of the radar image. The so called S-Mode PCA exploits the spatial dependence to compress


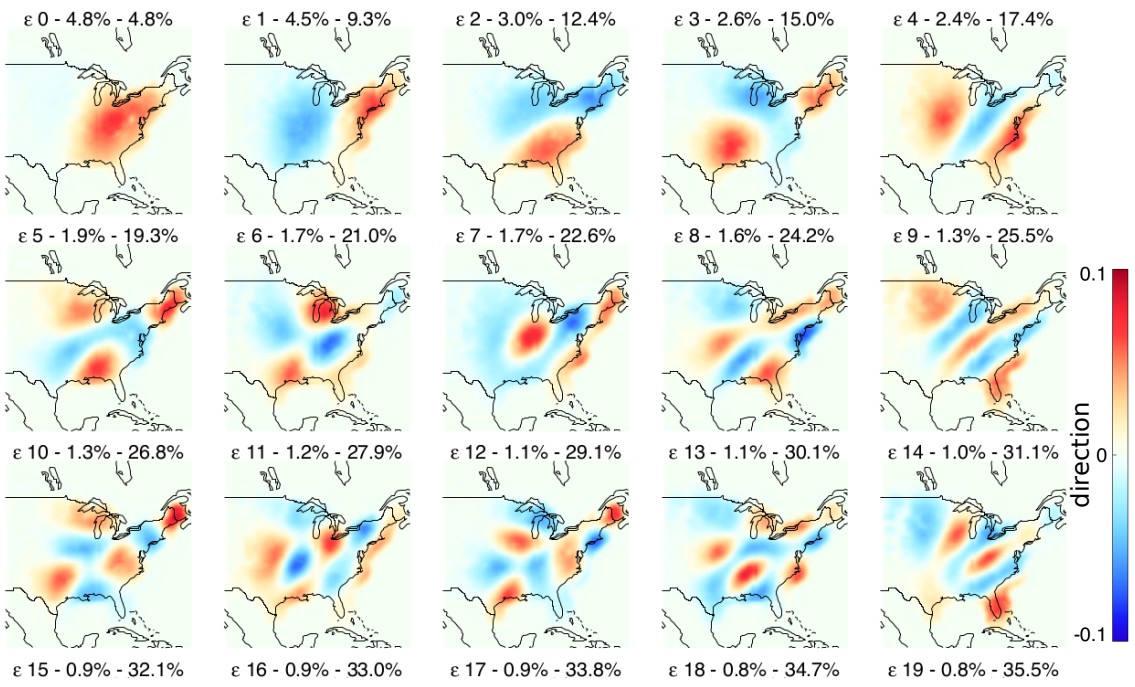

**Figure 10.** Eigenvectors (loadings) extracted by PCA from the US radar archive.

the information contained in the radar archive into a small set of principal components, which are here used as phase space
dimensions. The lower dimensional phase space is obtained by projecting the data matrix as follows:

$$\mathbf{Y}_{N,D} = \mathbf{X}_{N,M} \mathbf{U}_{M,D} \tag{5}$$

where $\mathbf{U}_{M,D}$ is the truncated matrix of eigenvectors (projection matrix), $\mathbf{X}_{N,M}$ is the original data matrix (Eq. 1), and $\mathbf{Y}_{N,D}$
is the (truncated) matrix of principal component scores (projected matrix), which contains the coordinates of radar images in
the phase space. For more details on the PCA implementation, we refer to Appendix B.

Due to the too large size of the US dataset, the radar fields were upscaled to a resolution of 64x64 $km^2$ pixels using a Haar
wavelet transform before applying PCA. This pre-processing step only marginally affects the search for analogues as we are
interested in similarity at synoptic scales.

## 4.2   Plotting phase space dimensions

Figure 10 shows the fields of eigenvectors computed from the US radar archive (1996-2016). The first eigenvector ($\epsilon$ 0) explains
only 4.8 % of the total variance and is characterized by positive values in the middle of the domain. It was found that the first
principal component (PC), associated to the first eigenvector, is strongly correlated with the field IMF (e.g. Foresti et al., 2015).
Radar images with precipitation located in this region will also have correspondingly high PC-0 scores. All other eigenvectors
can be interpreted in a similar way. For instance, the second ($\epsilon$ 1) and third ($\epsilon$ 2) eigenvectors discriminate precipitation patterns

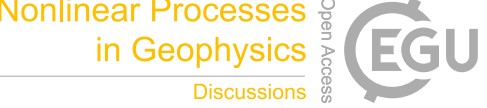

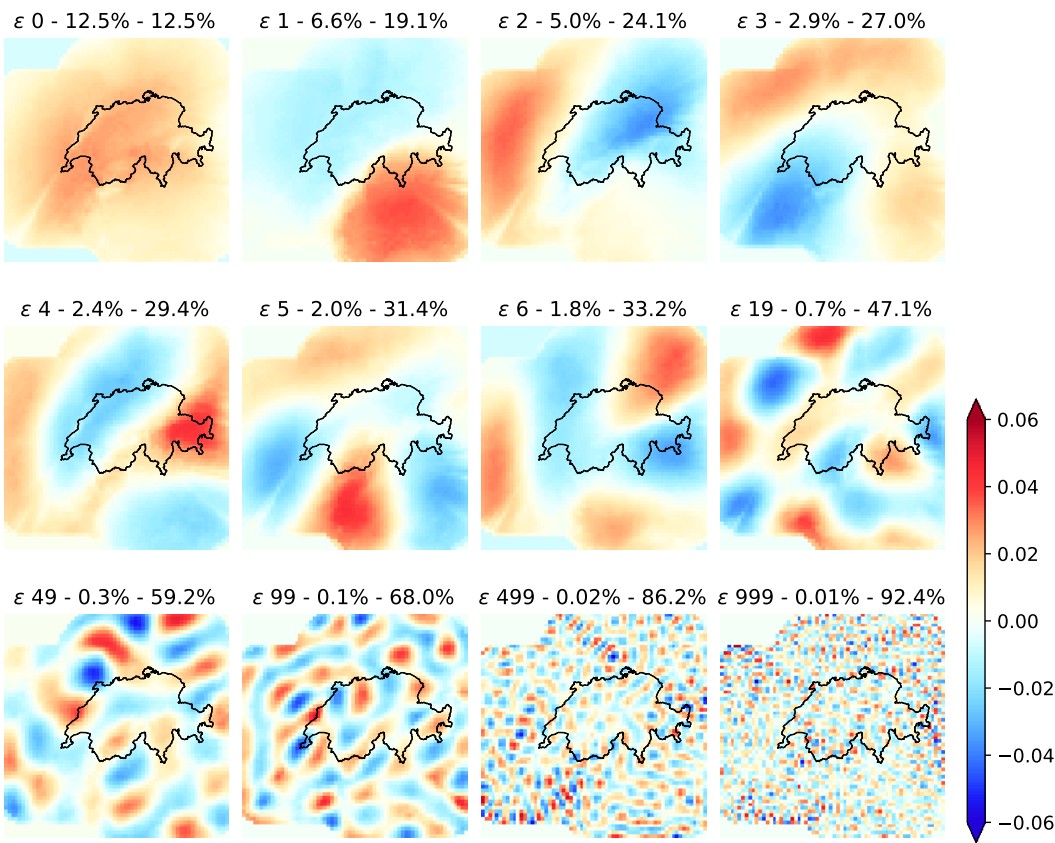

**Figure 11.** Eigenvectors (loadings) extracted by PCA from the Swiss radar archive.

in the West-East and North-South directions, respectively. Therefore, a precipitation system moving from West to East will
show an increase of PC-0 followed by an increase of PC-1. Finally, eigenvectors exhibit a characteristic sorting by spatial
scale.

Figure 11 shows the eigenvectors of the Swiss radar dataset (2005-2010), which also show the characteristic sorting by spatial
scale. The domain is much smaller compared with the US, but similar patterns can be observed, for example the tendency of
the first eigenvector to have high values in the middle of the domain, and the dipole-shape of the second and third eigenvectors
oriented in the North-South and East-West directions.

These shapes do not only highlight the most common precipitation regimes, but are also influenced by the rectangular shape
of the domain and the orthogonality constraints of PCA. These dipole effects are known in the literature as *Buell patterns* and
can complicate the meteorological interpretation of principal components (e.g. Richman, 1986).

In an attempt to improve their interpretation, we implemented a varimax rotation of the principal components (e.g. Richman,
1986), which were truncated at different thresholds of the cumulative explained variance (before rotation). The rotated eigen-





vectors highlighted some parts of the domain (with values close to zero elsewhere) and lost the sorting by spatial scale. As we did not find these results informative, they were not included in the paper.

Figure 11 also shows a few eigenvector fields for higher PC numbers (50, 100, 500 and 1000). After 500 components the eigenvectors become more noisy and describe very small scale precipitation features. Note that even after 500 eigenvectors

there is still almost 15% of unexplained variance.

PCA explains more variance with less components in the Swiss dataset. For instance, with 20 components the cumulative explained variances are 47.1% and 35.5% for the Swiss and US domains, respectively. This can be attributed to the smaller Swiss dataset, but also to the more frequent orographic precipitation events related to the presence of the Alps, which determines more predictable spatial patterns on the upwind and downwind sides of the Alpine chain.

A common pattern for both Swiss and US attractors is that PCA decomposes the dataset into a set of eigenvectors that represent decreasing spatial scales, similarly to what is obtained by a Fourier-based cascade decomposition of precipitation fields (Seed, 2003; Seed et al., 2013). This phenomenon is detailed further in the next section.

### 4.3    PCA vs Fourier analysis

The sinusoidal patterns of eigenvector fields are an outcome of the Toeplitz-like nature of the covariance matrix of spatially

correlated fields, whose eigenvectors represent sines and cosines of increasing frequencies. More precisely, for a stationary process the sinusoidal basis functions of the Fourier transform form a valid principal component basis, where the variance of each component represents the power spectrum (e.g. Simoncelli and Olshausen, 2001).

PCA derives the basis functions by decomposing an empirical covariance matrix. This may explain why in atmospheric science the principal components are referred to as *empirical orthogonal functions* (EOF Lorenz, 1956; Richman, 1986).

Instead, Fourier analysis assumes the orthogonal basis to be composed of sines and cosines, which is imposed prior to the analysis. Such assumptions simplify the use of Fourier analysis, which can be applied also to a single radar image. Instead, PCA needs an archive of radar images to derives the orthogonal basis. We find here again the inductive vs deductive dichotomy as in the definition of the phase space dimensions.

The similarity between PCA and Fourier decomposition creates interesting links to the cascade decomposition used in the

Short-Term Ensemble Prediction System (STEPS Seed, 2003; Bowler et al., 2006). In STEPS, the FFT is used to decompose and simulate precipitation fields within a multiplicative cascade framework, where each level represents precipitation features at different spatial scales.

Inspired by the relation to the cascade decomposition, Nerini et al. (2019) used PCA for blending radar ensemble nowcasts with NWP ensembles in a reduced space. An interesting future development arising from these findings could be to stochas-

tically simulate precipitation fields in the space of principal components, for example by extending the method of Link et al. (2019).





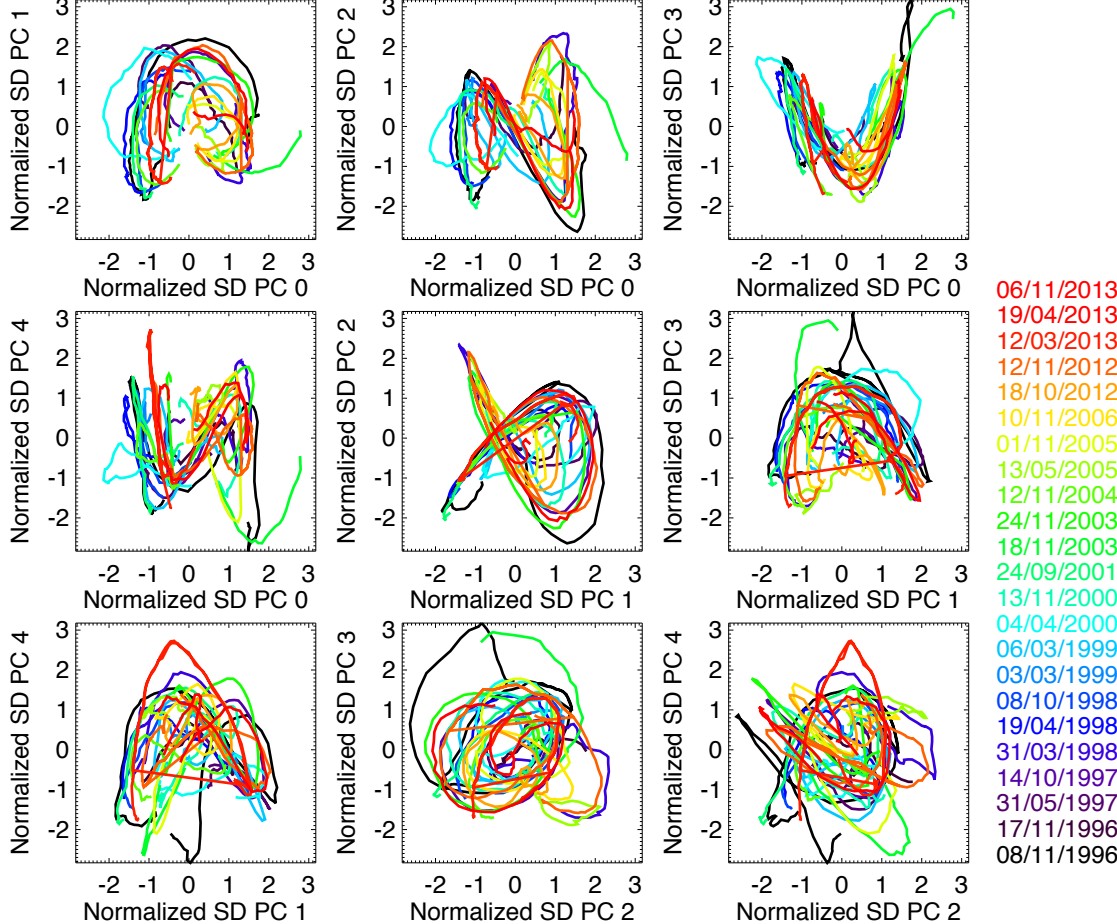

**Figure 12.** Trajectories of similar radar image sequences over the US in the space of standardized principal components. The dates of the 23 similar precipitation events are displayed on the right.

## 4.4 Phase space trajectories

Figure 12 shows the trajectories of US radar composite images in the space of first principal components (PCs). For this experiment, PCA was applied only to 23 manually selected similar events to better understand its inner workings.

An interesting observation is that PC trajectories define pseudo-regular patterns, which result from the translation of precipitation systems from the West to the East. The most regular and illustrative PC shapes are found in the three sub-panels located at the following (row, column): (1,2), (1,3), and (2,2). The one at (1,2) has a faint resemblance to the Lorenz attractor, which is only a fortunate coincidence.

The trajectories within the Swiss precipitation attractor also show patterns that are explained by the location and translation 375 of precipitation systems within the domain, as already observed by Foresti et al. (2015), but are not shown here.


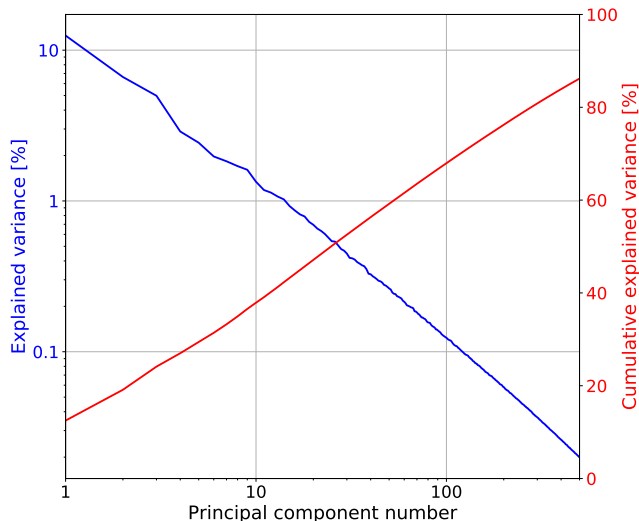

**Figure 13.** Explained variance and cumulative explained variance vs principal component number from the Swiss PCA.

## 4.5 Scaling properties

The sorting of eigenvectors by spatial scale observed in Section 4.2 is corroborated by plotting the explained and cumulative variance versus the ordinal principal component number in log-log scale, as shown in Fig. 13. Indeed, the explained variance draws a clear straight line (power law), similar to those obtained from Fourier-based scaling analyses (see e.g. Fig. 3).

The slow increase of cumulative explained variance does not allow to define a clear cutoff level to truncate the principal components. These results do not leave a lot of optimism concerning the definition of a low-dimensional attractor for precipitation based on PCA. Instead they point towards a stochastic approach for precipitation analysis and simulation.

One way to establish an empirical relation to Fourier-based scaling analysis is to convert the ordinal PC numbers of Fig. 13 into the corresponding spatial scales $\gamma$, which are represented by spatial wavelengths $\lambda = 1/k = 2\gamma$. For this task, we developed

the following methodology (see Fig. 14):

1. Compute the 2D Fourier spectra of the eigenvector fields (e.g. of Fig. 11),

2. Derive the 1D RAPS from the 2D spectra,

3. Estimate the most representative wavelength $\lambda$ from each 1D spectrum. We tested two methods:

    (a) *Maximum power method*, which returns the wavelength with maximum power,

(b) *Weighted average method*, which computes an average of wavelengths weighted by power.

4. Plot the obtained wavelength against PC number in log-log scale (Fig. 14a),



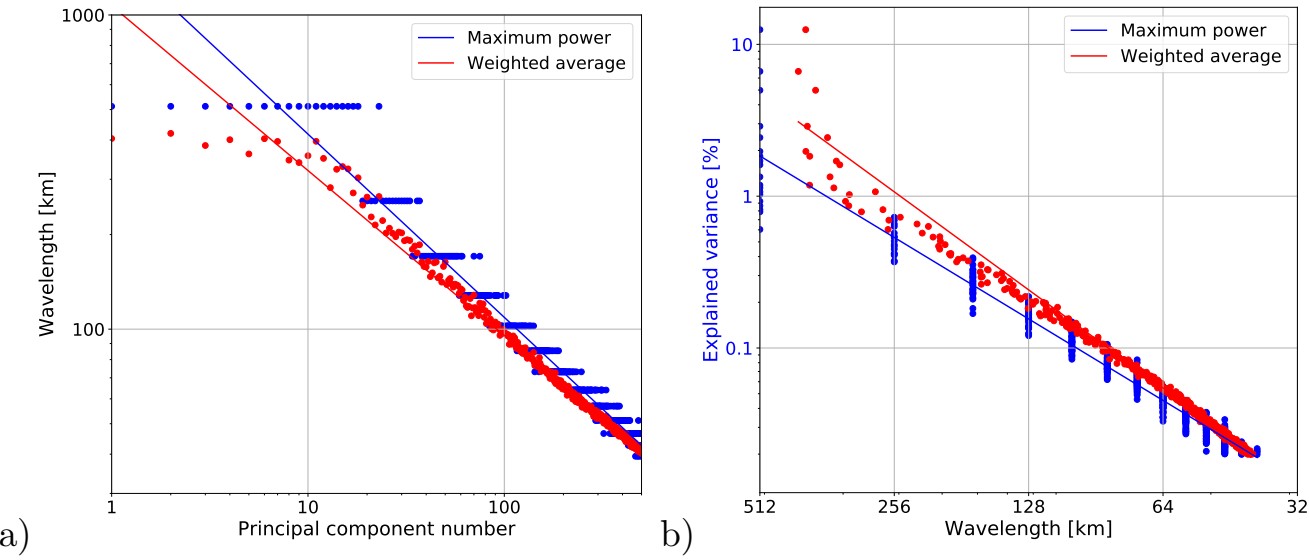

**Figure 14.** Derivation of spatial wavelength from PC number. a) Wavelength vs. PC number; b) Explained variance vs. wavelength (spatial frequency), which can be interpreted as a Fourier power spectrum.

5. Replace the PC number with the corresponding wavelength and plot it against the explained variance from Fig. 13 (Fig. 14b).

Figure 14 demonstrates the existence of a power-law relationship between the wavelength and both the PC number and explained variance. The results obtained by the maximum power and weighted average methods are quite similar, despite some deviations above the 300 km wavelength (due to wavelengths larger than the domain size).

These findings point out that there is no universal relationship that maps the ordinal PC number to the spatial scale, as the latter depends on the covariance matrix of a given dataset. However, the method proposed above offers a simple and effective way to reveal the spatial scale represented by a given eigenvector.

## 4.6 Fractal properties

Figure 15 shows an experiment to separate the predictable from the unpredictable precipitation scales using the Swiss attractor. The assumption is that the time series of PCs representing large scale features converge to a lower correlation dimension than the ones representing the more "stochastic" small scales. Note that a similar experiment was also done by Alberti et al. (2023) using multivariate empirical mode decomposition on the Lorenz system.

Figure 15a shows the temporal autocorrelation functions (ACFs) of a selection of principal component time series for each precipitation event. As expected, there is a decrease of the decorrelation time for increasing PCs. The very high correlations in the first hour, especially visible for PC1, could be related to the slow error growth in the 0-1 h range already observed in the





US attractor (see Fig. 8). However, they might also be explained by artifacts introduced by PCA, which artificially increase the smoothness of time series.

The first minimum of the ACF was used as time delay $\tau$ by the time-delay embedding method to estimate the associated correlation dimension of each PC time series (see Fig. 15b). The results show that all time series converge towards a finite correlation dimension, which grows from 2 to 4-5 when going from the $1^{st}$ to the $100^{th}$ PC. Only the last PC does not converge. These results highlight the expected underestimation of correlation dimension of the Grassberger-Procaccia method (Schertzer et al., 2001).

A major difficulty that we encountered in applying time-delay embedding is related to the short duration of precipitation events compared with the time delay $\tau$. In fact, if we consider a normal precipitation event lasting 24 hours (over the Swiss domain) and $\tau = 4$h, the maximum dimensionality of the embedding space is $D = \frac{24}{4} = 6$. Adding more dimensions will only include radar images with no precipitation to the time series and form a fixed point in the attractor, which adversely affects the estimation of the fractal dimension. This constraint is well visible in Fig . 15c, which shows the number of samples available

to compute the correlation dimension as a function of embedding dimension and PC number. One way to reduce this effect is by choosing larger domains to increase the probability there is precipitation somewhere.

Finally, even though the fractal dimensions estimates in this paper cannot be interpreted in absolute terms, they can be interpreted in relative terms, i.e. lower PCs exhibit stronger chaotic behaviour than larger PCs, which have a more stochastic behaviour.

## 4.7 Growth of errors

Figure 16 shows the forecast accuracy obtained by analogues for different PCA configurations and normalization of data. Each PCA configuration comprises a combination of $\text{Wav}_x$, $\text{PCA}_x$ and rotation parameters:

- $\text{Wav}_1$: before PCA; subtract mean from data columns,

- $\text{Wav}_2$: before PCA, subtract mean and divide data columns by standard deviation,

- $\text{PCA}_1$: after PCA, use raw PC scores,

- $\text{PCA}_2$: after PCA, subtract mean and divide PC scores by their standard deviation,

- $\text{PCA}_3$: after PCA, weight each PC score by the explained variance.

- Rotation (varimax): yes/no.

- Random: random selection of analogues.

The forecast quality of analogues was measured by three continuous verification scores, i.e. mean absolute deviation (MAD), root-mean square error (RMS), Pearson's correlation, one categorical score (critical success index, CSI), and two probabilistic scores, i.e. area under the ROC curve and Brier score at the 1 dBZ threshold. The verification was done using 50 precipitation




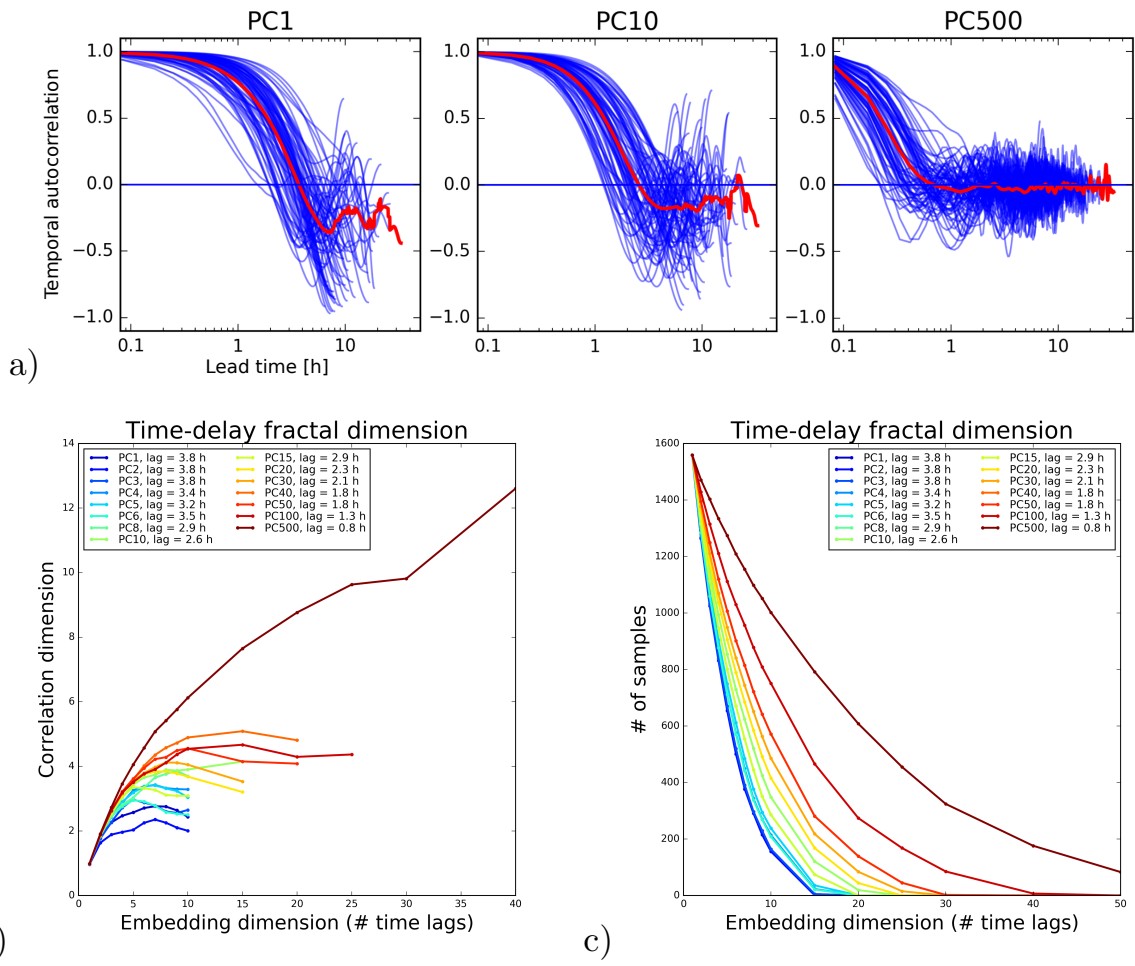

**Figure 15.** Correlation dimension analysis of principal component time series on the Swiss attractor. a) Temporal autocorrelation functions of the PC time series for different precipitation events (blue) and the average ACF (red); b) Correlation dimension vs embedding dimension for different PCs; c) Number of samples available for the estimation.



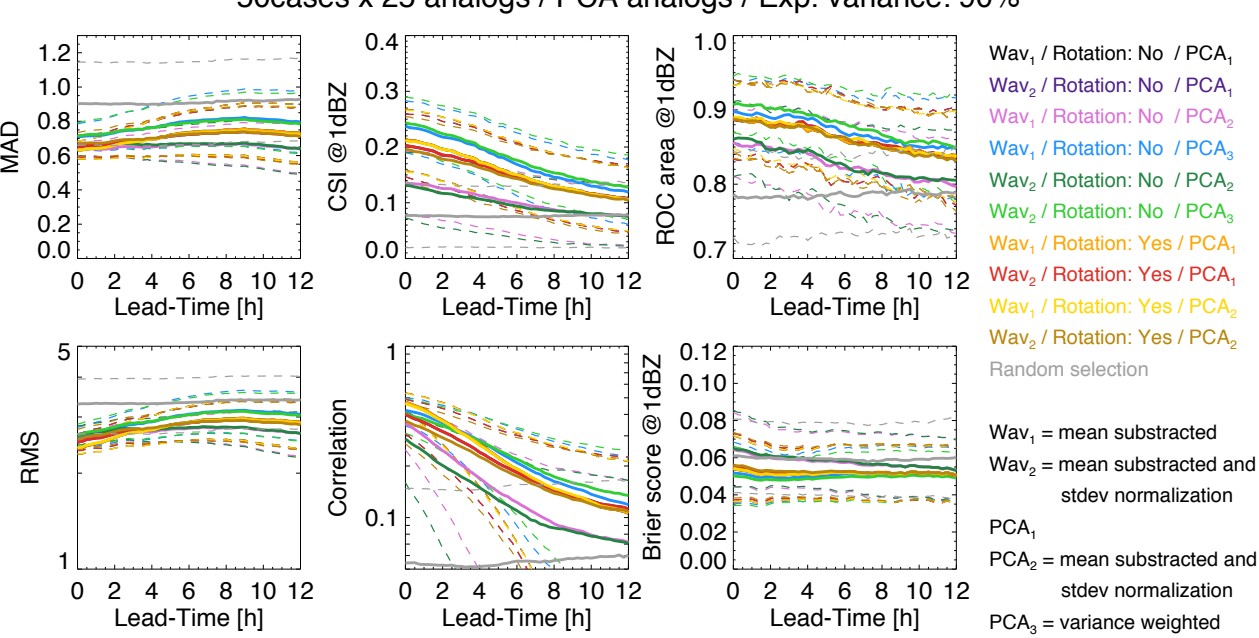

**Figure 16.** Testing different PCA settings for retrieving analogues in the US. The dashed lines show the $25^{th}$ and $75^{th}$ percentile values of the score distribution.

events in the US. For each of the 50 events 25 analogues were selected based on the smallest Euclidean distance in PC space and forcing them to be at least 16 hours from each other (for temporal independence). A 90% threshold of cumulative explained

variance was used to define the dimensionality of phase space.

According to MAD, RMS and correlation, the best configuration is $Wav_1$, Rotation (yes) and $PCA_2$. Instead, according to CSI, ROC and Brier score, the best configuration is $Wav_2$, Rotation (no) and $PCA_3$, but also $Wav_1$, Rotation (no) and $PCA_3$. In summary, it seems that rotating eigenvectors degrades the categorical scores but improves the continuous scores, i.e. the ability to predict more intense precipitation. Weighting the PCs by the explained variance improves the categorical scores at 1 dBZ

threshold, i.e. the ability to separate wet and dry areas. It is not yet clear how these conclusions generalize to different climatic regions.

This analysis highlights that one the attractor is defined it is not that simple to retrieve analogue states, i.e. practical implementation choices have an impact on the predictability estimations. Note that the low skill already at the start of the forecast (correlation $\approx$ 0.3-0.6 and CSI $\approx$ 0.15-0.25) is likely the result of verifying the forecast at the pixel resolution, while the re-

trieved analogues are only similar at large scales (see Sect. 2.2). Neighbourhood verification, e.g. using the fractions skill score (Roberts and Lean, 2008).





Similar to Foresti et al. (2015), we also searched analogues by minimizing the Euclidean distance of the last 2 (instead of 1) points in the PC space. Results were not surprising: the skill at the short lead times degraded and the one and longer lead times improved, but only slightly.

## 5   Conclusions

This paper explored a framework to construct empirical low dimensional precipitation attractors from multi-year archives of composite radar precipitation fields. The attractors were used to learn about the intrinsic predictability and various properties of precipitation fields. Data covering the Swiss Alps (2005-2010, $512 \times 512$ km$^2$ domain) and the continental US (1996-2016, $4096 \times 4096$ km$^2$ domain) were used.

We tested two approaches to define the attractor. The first approach uses as phase space dimensions selected domain-scale statistics of precipitations fields that are relevant for nowcasting applications, for example the precipitation fraction, mean precipitation and slopes of the Fourier power spectrum, which characterize the spatial autocorrelation. The second approach derives the phase space in a more objective way by principal component analysis, which also considers the location of precipitation.

After defining the phase space dimensions, we studied the fractal properties and error growth from analogues within both attractors. The pros and cons of the two types of attractors are summarized in Table 2. The main conclusions are:

- We could not find a unique objective way to define the phase space of the attractor, i.e. one is free to construct the attractor depending on the objective of the study or specific application.

- Graphical representation of the attractor as the density of points in various combinations of phase space dimensions provides useful insight into data dependencies and precipitation regimes (e.g. stratiform vs convective).

- The magnitude of the scaling break in radially-averaged power spectra of radar precipitation fields, previously observed by Gires et al. (2011) and Seed et al. (2013), is much more pronounced with isolated convective precipitation than stratiform precipitation.

- Error growth from analogues retrieved by using domain-scale statistics starts slow (0-1h, reason mostly unknown), continues fast (1-6h, unpredictable convective precipitation growth and decay), and slows down again before predictability is lost to a large extent (6h-20d, more predictable synoptic scales).

- The rate of error growth depends on the phase space used and initial location within the attractor.

- If the appropriate phase space dimensions are chosen, there is unexpectedly long intrinsic predictability of precipitation (several days), as shown with the US dataset.

- Predictability of domain-scale statistics is longer in the US than CH, which is attributed mostly to the larger domain, but also to the longer dataset.





**Table 2.** Advantages and disadvantages of the two types of attractors.

| | Deductive attractor: domain-scale statistics | Inductive attractor: principal components |
|---|---|---|
| Phase space | WAR, IMF, MM, $\beta_1$, $\beta_2$, etc | $PC_1$, $PC_2$, $\cdots$, $PC_D$ |
| Pros | - Phase space dimensions are interpretable.<br>- Phase space dimensions can be easily integrated into existing stochastic nowcasting systems. | - Phase space is extracted in an objective way.<br>- Phase space considers the location of precipitation.<br>- The decomposition accounts for scaling of variance and allows truncating the phase space at the desired level.<br>- Some domain-scale statistics are implicitly included. |
| Cons | - Analogues are only similar at the domain scale.<br>- The choice of phase space dimensions is subjective and depends on the application. | - Analogues are only similar at large scales.<br>- The PCA configuration and data normalization affects the quality of analogues. |

- By considering the spatial distribution of precipitation, PCA represents an useful framework for analysis, combination and simulation of precipitation fields.

- Fourier analysis can be used to derive the spatial scales corresponding to eigenvector fields extracted by PCA.

- The explained variance by PCA is scaling with both the ordinal PC number and corresponding spatial scale, which has a clear connection to Fourier-based decomposition of precipitation fields (e.g. Seed, 2003; Bowler et al., 2006).

- Fractal analysis of the principal component times series reveals that low PCs have a stronger chaotic contribution than high PCs, which have a stronger stochastic component.

The application of tools used in chaos theory, such as time-delay embedding and the correlation dimension method, is 490 complicated by the precipitation intermittency, finite event duration, non-gaussian distribution and multifractal properties. These difficulties are also reflected in the analysis of derived phase space variables (MM, WAR, PCs, etc). In addition, the validity of theorems and assumptions from chaos theory are pushed beyond their limits because precipitation is not only the result of dynamical processes, but also of (stochastic) microphysical processes. It is important to mention that the current study did not have the ambition to demonstrate that the precipitation attractor is of (finite) low dimensionality (see discussion in 495 Appendix D), but only to gain additional insight by testing different approaches.

Future perspectives comprise both improvements of the methodology and more practical applications. The methodology can be improved by integrating for example faster analogue retrieval methods (e.g. Franch et al., 2019) or more robust methods for estimating fractal dimensions (e.g. Golay and Kanevski, 2015; Camastra and Staiano, 2016; Pons et al., 2023). The size of the dataset could also be extended, although the main conclusions are not expected to change.

Concerning possible applications, it is not yet clear how to exploit the gathered knowledge to improve precipitation forecasting in practice. For instance, both NWP forecasting and stochastic nowcasting methods are known to underestimate the





forecast uncertainty, i.e. the ensembles are underdispersive. One possibility would be to drive stochastic simulations with the large scale features given by analogues. Another possibility could be to seamlessly blend forecast probabilities derived from extrapolation nowcasts, NWP models and analogues.

Finally, a completely different methodology, which has attracted the attention in the atmospheric science community for quite some time, is to train machine learning algorithms to optimally extract the localized predictable patterns from the data (Foresti et al., 2018, 2019). It could be insightful to use the methodology presented in this paper to understand what was exactly learned by machine learning algorithms in terms of predictability.

## Appendix A: Fourier analysis of precipitation fields

The discrete 2D power spectrum is defined as the squared norm of the complex Fourier transform:

$$P(k_x, k_y) = \frac{1}{M} |\mathcal{F}\{\mathbf{Z} - \overline{\mathbf{Z}}\}(k_x, k_y)|^2, \tag{A1}$$

where $M$ is the number of image pixels, $\mathbf{Z}$ the precipitation field, $\overline{\mathbf{Z}}$ the mean precipitation of the field, $\mathcal{F}$ the fast Fourier transform operator, and $(k_x, k_y)$ the wave numbers (corresponding to spatial frequencies). The 2D spectrum informs about the distribution of variance with spatial frequency and is a useful tool to analyze and model the spatial structure of rainfall fields

(e.g. Seed, 2003; Nerini et al., 2017).

The spatial autocorrelation function (ACF) is obtained via the Wiener-Khintchin theorem as the inverse Fourier transform of the power spectrum under the assumption of stationarity (e.g. Nerini et al., 2017; Jameson et al., 2018):

$$P'(x, y) = \frac{1}{Var\{\mathbf{Z}\}} \mathcal{F}^{-1}\{P(k_x, k_y)\}. \tag{A2}$$

where $Var\{\mathbf{Z}\}$ is the precipitation field variance. Since the autocorrelation and the spectrum form a Fourier transform pair,

they both convey the same information, the former in physical space and the latter in the space of frequencies.

By assuming isotropy, from the 2D power spectrum we can derive a radially-averaged 1D spectrum (RAPS):

$$P(|k|) = \frac{1}{|Z|} \sum_{z=1}^{|Z|} P(Z_z), \tag{A3}$$

where $Z = \{(k_x, k_y)_1, \ldots, (k_x, k_y)_{|Z|}\}$ is the set of wave numbers for which $|k| \leq \sqrt{k_x^2 + k_y^2} < |k| + 1$. The same can be done for the 2D spatial autocorrelation.

## Appendix B: Principal component analysis

Starting from the log transformed precipitation data matrix $\mathbf{X}_{N,M}$, PCA consists of the following steps:

1. Center the data matrix by the column means, i.e.
   $$\hat{\mathbf{X}}_{N,M} = \mathbf{X}_{N,M} - \mathbf{1}\overline{\mathbf{x}}^{\mathbf{T}}.$$





2. Compute the covariance matrix to estimate the linear dependence of variables, i.e. $\mathbf{C}_{M,M} = \hat{\mathbf{X}}_{N,M}^T \hat{\mathbf{X}}_{N,M}$.

3. Diagonalize $\mathbf{C}_{M,M}$ by eigenvalue decomposition (EVD), i.e. $\mathbf{C} = \mathbf{U}\mathbf{V}\mathbf{U}^T$, where $\mathbf{U}_{M,M}$ is the orthogonal matrix of eigenvectors (each column is one vector) and $\mathbf{V}_{M,M}$ is the diagonal matrix of eigenvalues $v_i$.

4. (Optional) Rotate the eigenvectors to enhance interpretation (e.g. Richman, 1986), e.g. using Varimax.

5. Project the original data matrix into the space spanned by eigenvectors, i.e. $\mathbf{Y}_{N,M} = \hat{\mathbf{X}}_{N,M}\mathbf{U}_{M,M}$.

Eigenvectors are sorted by decreasing amount of explained variance, which can be truncated such that $D \ll M$.

An alternative way to perform PCA is by singular value decomposition (SVD) of the data matrix (e.g. Jolliffe, 2002). SVD factorization of the centered data matrix is obtained as:

$$\hat{\mathbf{X}}_{N,M} = \mathbf{L}_{N,N}\mathbf{S}_{N,M}\mathbf{R}_{M,M}^T \tag{B1}$$

where $\mathbf{L}$ is the matrix of left singular vectors, $\mathbf{R}$ the matrix of right singular vectors, and $\mathbf{S}$ the diagonal matrix of singular values $s_i$. The eigenvalues can be calculated from the singular values as $v_i = s_i^2$.

Since SVD does not require the computation of the covariance matrix, it has larger numerical stability than EVD. However, SVD is slower than EVD if $N \gg M$. In such case, one can perform a reduced SVD to avoid storing the large matrix $\mathbf{L}_{N,N}$. Finally, the projected data matrix is computed as $\mathbf{Y}_{N,M} = \mathbf{L}_{N,N}\mathbf{S}_{N,M} = \mathbf{X}_{N,M}\mathbf{R}_{M,M}$.

For the Swiss archive, we used the SVD-based PCA decomposition available in the Python library sklearn (Pedregosa et al., 2011). For the US archive, we used the classical covariance-based PCA decomposition written in IDL.

**Appendix C: Time-delay embedding**

An important concept for studying nonlinear dynamical systems is the *time-delay embedding* theorem (Takens, 1981). Takens's theorem defines the conditions for which the dynamics of a smooth attractor can be reconstructed from a time series of observations of a single state space variable. Note that the assumption of smoothness may not apply beyond a certain data noise level (e.g. Schertzer et al., 2001).

Takens' theorem is applied by lagging multiple times the time series of a state-space variable:

$$\mathbf{X}_{N,D} = [\mathbf{x}_t, \mathbf{x}_{t-\tau}, \cdots, \mathbf{x}_{t-D\tau}], \tag{C1}$$

where $\mathbf{x}_t$ is the original time series, $\mathbf{x}_{t-\tau}$ is the time series delayed by $\tau$, and $D$ is the dimensionality of embedding space. In this paper, we have chosen $\tau$ to be equal to the first minimum of the temporal autocorrelation function.

**Appendix D: Correlation dimension method**

The correlation dimension method, known as Grassberger-Procaccia algorithm, estimates the fractal dimension of an attractor by counting the number of points that are contained in a $D$-dimensional sphere of increasing radius $r$, i.e. the correlation





integral (Grassberger, 1986):

$$C(r) = \frac{1}{N^2} \sum_{\forall i} H(r - \Delta r) \tag{D1}$$

where $H$ is the heaviside function. The counting is performed for each point of the attractor. A log-log plot of $C(r)$ versus $r$

gives an estimation of the fractal dimension:

$$C(\epsilon) \propto r^f, \tag{D2}$$
$$log\big(C(\epsilon)\big) \propto c + f \log(r), \tag{D3}$$

where $c$ is an offset and $f$ is the fractal dimension (see Fig. D1 for an example using US data). In a 2D embedding space, randomly distributed points would give an $f \approx 2$, while points along a straight line would give $f \approx 1$. In this paper, the terms

fractal and correlation dimension are used interchangeably as measures of the intrinsic dimensionality of a dataset.

The Grassberger-Procaccia algorithm is known to underestimate the fractal dimension, which was the subject of controversial discussions questioning claims about the existence of low-dimensional attractors of atmospheric and hydrological processes (see e.g. Grassberger, 1986; Lorenz, 1991; Koutsoyiannis and Pachakis, 1996; Sivakumar et al., 2001a; Schertzer et al., 2001; Sivakumar et al., 2001b; Koutsoyiannis, 2006). As a consequence, new techniques are being developed to overcome its

limitations (e.g. Golay and Kanevski, 2015; Camastra and Staiano, 2016).

**Appendix E:  CONUS coverage**

*Author contributions.*  Loris Foresti performed the analyses using the Swiss radar dataset and wrote the paper. Bernat Puigdomènech Treserras and Aitor Atencia performed the analyses using the US radar dataset. Daniele Nerini supported the implementation and interpretation of results. Marco Gabella, Ioannis Sideris and Urs Germann supported the study with numerous discussions about radar precipitation, nowcast-

575 ing and chaos theory. Isztar Zawadzki originated the idea of precipitation attractor and supervised the study with unparalleled enthusiasm and creativity for as long as he could.

*Competing interests.*  The authors have no competing interests.

*Acknowledgements.*  This study was supported by the Swiss National Science Foundation Ambizione project *Precipitation attractor from radar and satellite data archives and implications for seamless very short-term forecasting* (PZ00P2_161316). Lea Beusch is thanked for

reviewing an earlier version of the paper.

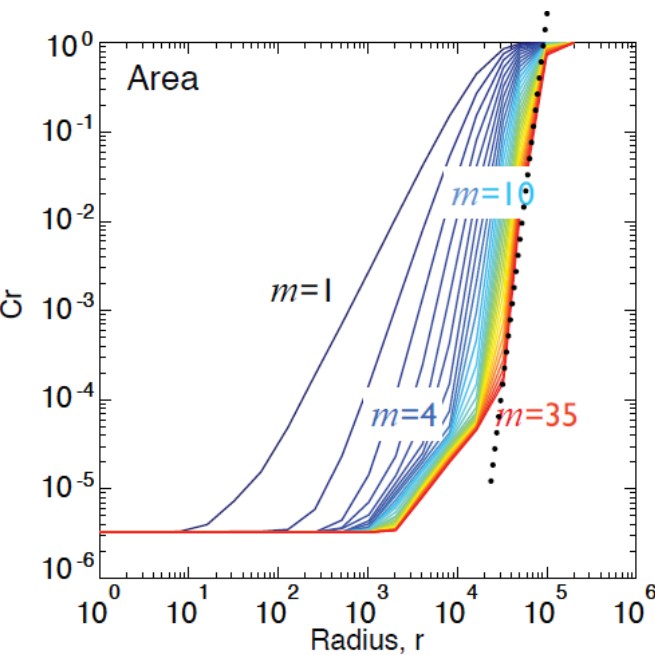

**Figure D1.** Example estimation of correlation dimension by finding the maximum slope in a log-log plot of correlation integral $C(r)$ vs search radius $r$. This example uses time-delay embedding on the time series of radar precipitation area on the US domain.

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





## NEXRAD - TDWR Coverage Below 10,000 Feet AGL

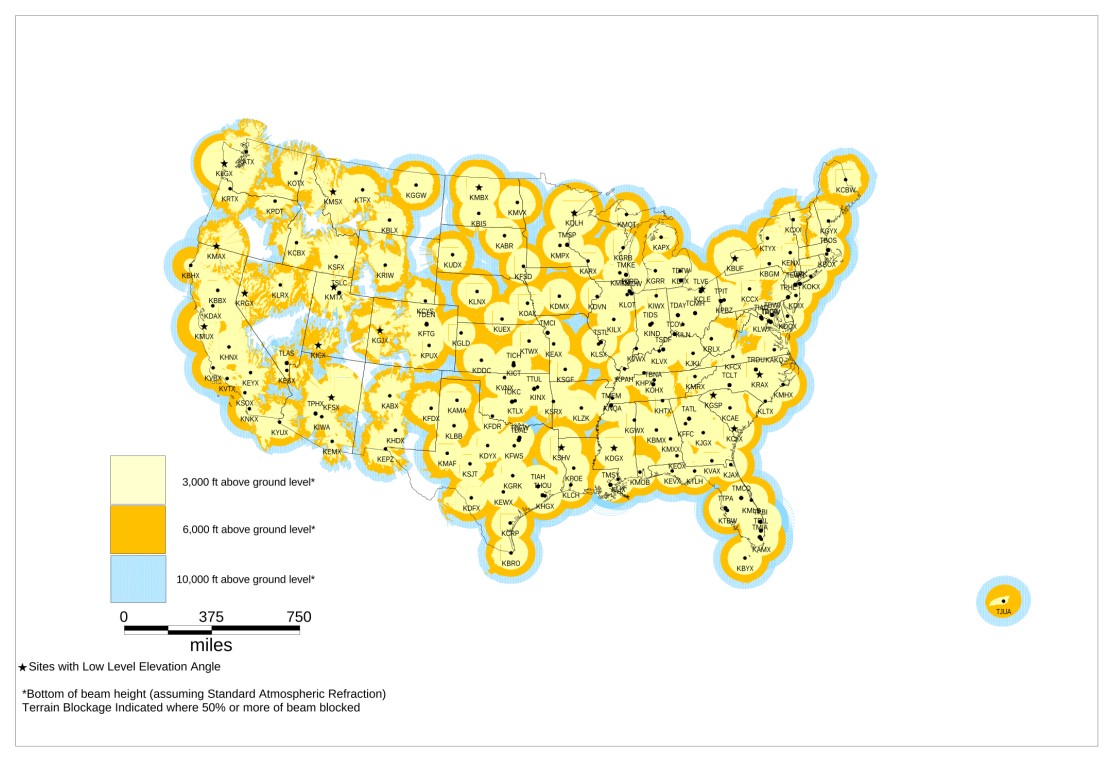

**Figure E1.** Weather radar coverage in the continental US. Downloaded from https://www.roc.noaa.gov/WSR88D/Maps.aspx on 03.09.2023.

De Cruz, L., Schubert, S., Demaeyer, J., Lucarini, V., and Vannitsem, S.: Exploring the Lyapunov instability properties of high-dimensional atmospheric and climate models, Nonlinear Processes in Geophysics, 25, 387–412, https://doi.org/10.5194/npg-25-387-2018, 2018.

Fabry, F., Meunier, V., Treserras, B., Cournoyer, A., and Nelson, B.: On the climatological use of radar data mosaics: possibilities and challenges, Bull. Amer. Meteor. Soc., 98, 2135–2148, 2017.

Foresti, L., Panziera, L., Mandapaka, P. V., Germann, U., and Seed, A.: Retrieval of analogue radar images for ensemble nowcasting of orographic rainfall, Meteorol. Appl., 22, 141–155, 2015.

Foresti, L., Sideris, I. V., Panziera, L., Nerini, D., and Germann, U.: A 10-year radar-based analysis of orographic precipitation

growth and decay patterns over the Swiss Alpine region, Quarterly Journal of the Royal Meteorological Society, 144, 2277–2301, https://doi.org/https://doi.org/10.1002/qj.3364, 2018.

Foresti, L., Sideris, I. V., Nerini, D., Beusch, L., and Germann, U.: Using a 10-year radar archive for nowcasting precipitation growth and decay - a probabilistic machine learning approach, Weather and Forecasting, 34, 1547–1569, https://doi.org/10.1175/WAF-D-18-0206.1, 2019.

Franch, G., Jurman, G., Coviello, L., Pendesini, M., and Furlanello, C.: MASS-UMAP: Fast and Accurate Analog Ensemble Search in Weather Radar Archives, Remote Sens., 11, 2922, 2019.





Germann, U., Galli, G., Boscacci, M., and Bolliger, M.: Radar precipitation measurement in a mountainous region, Q. J. R. Meteorol. Soc.,
132, 1669–1692, 2006a.

Germann, U., Zawadzki, I., and Turner, B.: Predictability of Precipitation from Continental Radar Images. Part IV: Limits to Prediction,
Journal of the Atmospheric Sciences, 63, 2092 – 2108, https://doi.org/10.1175/JAS3735.1, 2006b.

Germann, U., Boscacci, M., Clementi, L., Gabella, M., Hering, A., Sartori, M., Sideris, I. V., and Calpini, B.: Weather Radar in Complex
Orography, Remote Sensing, 14, https://doi.org/10.3390/rs14030503, 2022.

Gires, A., Tchiguirinskaia, I., Schertzer, D., and Lovejoy, S.: Multifractal and spatio-temporal analysis of the rainfall output of the Meso-NH
model and radar data, Hydrological Sciences Journal, 56, 380–396, https://doi.org/10.1080/02626667.2011.564174, 2011.

Golay, J. and Kanevski, M.: A new estimator of intrinsic dimension based on the multipoint Morisita index, Pattern Recognition, 48, 4070–
4081, 2015.

Grassberger, P.: Do climatic attractors exist?, Letters to Nature, 323, 609–611, 1986.

Grassberger, P. and Procaccia, I.: Measuring the strangeness of strange attractors, Physica D: Nonlinear Phenomena, 9, 189–208, 1983.

Houze, R.: Cloud Dynamics, Academic Press, 2014.

Jameson, A., Larsen, M., and Kostinski, A.: On the Detection of Statistical Heterogeneity in Rain Measurements, J. Atmos. Oceanic Technol.,
35, 1399–1413, 2018.

Jolliffe, I.: Principal Component Analysis ($2^{nd}$ Ed.), Springer, 2002.

Kantz, H. and Schreiber, T.: Nonlinear Time Series Analysis ($2^{nd}$ Ed.), Cambridge University Press, 2004.

Koutsoyiannis, D.: On the quest of chaotic attractors in hydrological processes, Hydrological Sciences, 51, 1065–1091, 2006.

Koutsoyiannis, D. and Pachakis, D.: Deterministic chaos versus stochasticity in analysis and modeling of point rainfall series, Journal of
Geophysical Research-Atmospheres, 101 (D21), 26 441–26 451, 1996.

Li, J. and Ding, R.: Temporal-spatial distribution of atmospheric predictability limit by local dynamical analogues, Mon. Wea. Rev., 139,
3265–3283, 2011.

Lichtenberg, A. J. and Lieberman, M. A.: Regular and Chaotic Dynamics, no. 38 in Applied Mathematical Sciences, Springer-Verlag, New
York, NY, 2nd edn., 1992.

Link, R., Snyder, A., Lynch, C., Hartin, C., Kravitz, B., and Bond-Lamberty, B.: Fldgen v1.0: an emulator with internal variability and space–
time correlation for Earth system models, Geoscientific Model Development, 12, 1477–1489, https://doi.org/10.5194/gmd-12-1477-2019,
2019.

Lorenz, E. N.: Empirical orthogonal functions and statistical weather prediction, Tech. Rep. Scientific Report No. 1, Statistical Forecasting
Project, Air Force Research Laboratories, Office of Aerospace Research, USAF, Bedford, MA, 1956.

Lorenz, E. N.: Deterministic nonperiodic flow, J. Atmos. Sci., 20, 130–141, 1963.

Lorenz, E. N.: Atmospheric predictability as revealed by naturally occurring analogues, J. Atmos. Sci., 26, 636–646, 1969.

Lorenz, E. N.: Dimensions of weather and climate attractors, Letters to Nature, 353, 241–244, 1991.

Lorenz, E. N.: Predictability - a problem partly solved, in: Proc. Seminar on Predictability, Volume 1. European Centre for Medium-Range
Weather Forecast, Shinfield Park, Reading, Berkshire, United Kingdom., 1996.

Lovejoy, S. and Schertzer, D.: The Weather and Climate: Emergent Laws and Multifractal Cascades, Cambridge University Press, 2013.

Menabde, M.: Bounded lognormal cascades as quasi-multiaffine random processes, Nonlinear Processes in Geophysics, 5, 63–68, 1998.

Nerini, D., Besic, N., Sideris, I., Germann, U., and Foresti, L.: A non-stationary stochastic ensemble generator for radar rainfall fields based
on the short-space Fourier transform, Hydrol. Earth Syst. Sci., 21, 2777–2797, 2017.



Nerini, D., Foresti, L., Leuenberger, D., Robert, S., and Germann, U.: A reduced-space ensemble Kalman filter approach for flow-dependent integration of radar extrapolation nowcasts and NWP precipitation ensembles, Mon. Wea. Rev., 147, 987 – 1006, https://doi.org/10.1175/MWR-D-18-0258.1, 2019.

Nicolis, C., Vannitsem, S., and Royer, J.-F.: Short-range predictability of the atmosphere: Mechanisms for superexponential error growth, Quart. Roy. Meteor. Soc., 121, 705–722, 1995.

Palmer, T. and Hagedorn, R., eds.: Predictability of Weather and Climate, Cambridge University Press, 2006.

Pedregosa, F., Varoquaux, G., Gramfort, A., Michel, V., Thirion, B., Grisel, O., Blondel, M., Prettenhofer, P., Weiss, R., Dubourg, V., Vanderplas, J., Passos, A., Cournapeau, D., Brucher, M., Perrot, M., and Duchesnay, E.: Scikit-learn: Machine Learning in Python, Journal of Machine Learning Research, 12, 2825–2830, 2011.

Pegram, G. G. S. and Clothier, A. N.: High resolution space-time modelling of rainfall: The "String of Beads" model, J. Hydrol., 241, 26–41, 660  2001.

Pons, F., Messori, G., and Faranda, D.: Statistical performance of local attractor dimension estimators in non-Axiom A dynamical systems, Chaos: An Interdisciplinary Journal of Nonlinear Science, 33, 073 143, https://doi.org/10.1063/5.0152370, 2023.

Pulkkinen, S., Nerini, D., Pérez-Hortal, A., Velasco-Forero, C., Seed, A., Germann, U., and Foresti, L.: Pysteps: An open-source python library for probabilistic precipitation nowcasting (v1.0), Geoscientific Model Development, 12, 4185–4219, 2019.

Richman, M. B.: Review article: Rotation of principal components, J. Climatology, 6, 293–335, 1986.

Roberts, N. M. and Lean, H. W.: Scale-Selective Verification of Rainfall Accumulations from High-Resolution Forecasts of Convective Events, Monthly Weather Review, 136, 78 – 97, https://doi.org/https://doi.org/10.1175/2007MWR2123.1, 2008.

Schertzer, D. and Lovejoy, S.: Physical modeling and analysis of rain and clouds by anisotropic scaling multiplicative processes, J. Geophys. Res., 92(D8), 9693–9714, 1987.

Schertzer, D., Tchiguirinskaia, I., Lovejoy, S., Hubert, P., Bendjoudi, H., and Larchevêque, M.: Which chaos in the rainfall–runoff process? Disussion of "Evidence of chaos in the rainfall-runoff process", Hydrological Sciences Journal, 47, 139–148, 2001.

Schiemann, R., Liniger, M., and Frei, C.: Reduced space optimal interpolation of daily rain gauge precipitation in Switzerland, J. Geophys. Res., 115, D14 109, 2010.

Seed, A.: A dynamic and spatial scaling approach to advection forecasting, J. Appl. Meteorol., 42, 381–388, 2003.

Seed, A., Pierce, C. E., and Norman, K.: Formulation and evaluation of a scale decomposition-based stochastic precipitation nowcast scheme, Water Resour. Res., 49, 6624–6641, 2013.

Sideris, I.: Measure of orbital stickiness and chaos strength, Phys. Rev. E, https://doi.org/https://doi.org/10.1103/PhysRevE.73.066217, 2006.

Sideris, I. V., Foresti, L., Nerini, D., and Germann, U.: NowPrecip: localized precipitation nowcasting in the complex terrain of Switzerland, Quarterly Journal of the Royal Meteorological Society, 146, 1768–1800, https://doi.org/https://doi.org/10.1002/qj.3766, 2020.

Simoncelli, E. and Olshausen, B.: Natural image statistics and neural representation, Annu. Rev. Neurosci., 4, 1193–2216, 2001.

Sivakumar, B., Berndtsson, R., Olsson, J., and Jinno, K.: Reply to "Which chaos in the rainfall-runoff process", Hydrological Sciences Journal, 47, 149–158, 2001a.

Sivakumar, B., Berndtsson, R., Olsson, J., and Jinno, K.: Evidence of chaos in the rainfall-runoff process, Hydrological Sciences Journal, 46, 131–145, 2001b.

Surcel, M., Zawadzki, I., and Yau, M. K.: A study on the scale dependence of the predictability of precipitation patterns, J. Atmos. Sci., 72, 216–235, 2015.





Takens, F.: Dynamical Systems and Turbulence, Lecture Notes in Mathematics, vol. 898, chap. Detecting strange attractors in turbulence, pp. 366–381, Springer, 1981.

Toth, Z.: Estimation of atmospheric predictability by circulation analogues, Mon. Wea. Rev., 119, 65–72, 1991.

Van Den Dool, H. M.: Searching for analogs: how long must we wait?, Tellus A, 46, 314–324, 1994.

Villarini, G. and Krajewski, W.: Review of the Different Sources of Uncertainty in Single Polarization Radar-Based Estimates of Rainfall, Surv. Geophys., 31, 107–129, https://doi.org/10.1007/s10712-009-9079-x, 2010.