# Peer review of "A quest for precipitation attractors in weather radar archives"

_Nonlinear Processes in Geophysics, 2023_

## Author Response (AR1)

**Reviewer 1 (Citation: https://doi.org/10.5194/npg-2023-24-RC1)**

Summary
The paper describes a "quest for precipitation attractors in weather radar archives". The topic is interesting and relevant for NPG. I have to admit that the title describes very well the content of the paper, which is indeed a quest that sometimes gets the reader a bit lost with regards to its purpose. I mean, in general, each step seems to be properly done, is well explained (below you can find a list of rather minor comments / suggestions to help the reader) and has some interests.
What is missing is a kind of overall picture / goal of the paper; or discussion about what it brings to rainfall understanding or potential applications which are only briefly addressed. I should also stress that the differences in methodologies for US and Swiss data do not make understanding of the overall purpose easier. In addition, not all plots are discussed in the text.
Therefore, I invite authors to revise their paper before it can be published. Modifications required should aim at helping the reader grasp what the obtained results bring to the understanding of rainfall and/or to potential practical applications.
Answer from the authors:
We appreciate a lot your very useful comments, which allowed us to substantially improve the manuscript. It was indeed not easy to define a storyline for the quest of the attractor using relatively different methods & metrics. This is also due to the fact that less promising methods in one attractor were not replicated onto the other, where different experiments were attempted. Therefore, we fully agree and expanded the explanations to help the reader better understand the purpose of the studies and potential applications (e.g. the attractor could be used also for radar quality control). We also regenerated the error growth plot of the Swiss attractor to use the same scaling and error metric of the US attractor.

Question from reviewer:
- l. 40-43: indeed, since methodologies are not the same and there is always a switch between the two case studies; it makes it a bit hard for the reader to understand how to compare the results and how all this makes sense. A good illustration is the use of different metrics (Eq. 2 or 3) to quantify error growth, or the sentence l. 252-253, or the fact that Fig. 8 and 9 are in log and log-log respectively?
Answer from the authors:
We fully agree with this comment and have revised the paper as follows.
  1) Different error growth metrics: In the revised version of the paper, we used the standard deviation as measure of error growth for both the US and Swiss attractors and updated Fig 9 (the plot using the robust measure of spread was moved to a new Appendix for completeness). Unfortunately, we did not manage to homogenize the methods to retrieve analogues and those defining the loss of

predictability (the kernel regression). As the error growth metrics are standardized by their sample climatological value, we expect the qualitative interpretations of predictability estimates to remain similar.

2) Different fractal dimensions analyses: In the revised manuscript, we rephrased the sentence at l. 252-253 to explain why this analysis was not replicated in the Swiss attractor. Instead of trying to interpret fractal dimension values in absolute terms, in the Swiss attractor we exploited PCA to estimate scale-dependent fractal dimensions, which can be compared in relative terms.

3) Different scaling of error growth axis (log-lin vs log-log): We fully agree that it is not ideal to compare two figures with different scaling. Therefore, we regenerated Fig. 9 (Swiss attractor) using the same scaling as Fig 8. (US attractor), i.e. using a log time and linear error to highlight the longer range of predictability. In addition, as required by reviewer 2, we also generated a figure using a standard linear time vs log error scaling, which was added into a new Appendix.

We also expanded the introductory chapter (l. 41-43) to explain that less successful experiments in one attractor were abandoned and not replicated onto the other, where we explored more promising directions. Also, it is not the objective of the paper to derive values that can be taken at face value, but rather to gain some qualitative insights.

Question from reviewer:
- Fig. 1: part of the domain is not covered by radar data. How are these missing values handled in the analysis?
Answer from the authors:
This is an interesting question as the handling of no data and zero precipitation could affect the analyses. For PCA, we set all values outside the domain to zero (we specified this in the revised version in Appendix B). For Fourier analyses, we set both the no data and zero precipitation to the minimum precipitation thresholds (l. 171-174). The remaining global statistics (WAR, IMF, etc) were normalized by the number of pixels over the radar domain (we specified this where missing in Sect. 3.1).

Question from reviewer:
- Section 2.2 is quite hard to follow and more explanations would be helpful to understand better the interpretations suggested by the authors.
Answer from the authors:
We expanded the section to provide more explanations. The methodology is easier to understand in the revised version.

Question from reviewer:
- l. 153-154: why using only the fraction of precipitation which does not include information on its spatial distribution, and not the fractal dimension on the rainfall

support for example to remain in the scale invariant framework sometimes addressed in the paper. Are results sensitive to the threshold defining rainy and dry pixels?
Answer from the authors:
We do not fully understand this comment. In fact, the fractal dimension of the rainfall support is a single number describing a global property of the field as is the fraction of precipitation. In this paper, to remain in the scale-invariant framework, we used the spectral exponent β (describing rainfall fluctuations) instead of the fractal dimension of its support (rain vs no rain). There is a relation between these two quantities (see e.g. Bies et al. 2016, https://doi.org/10.3390/sym8070066). The spatial distribution of precipitation is considered by the PCA framework, which is described in Section 4. In addition, it is perfectly possible to condition the stochastic fields on the spatially-localized fraction of precipitation, mean precipitation and Fourier transform (see e.g. Nerini et al., 2017; Sideris et al., 2020). We added one sentence in the paper to bring the attention of the reader to this possibility.

Question from reviewer:
- l. 188-189: how is it computed? For each pixel and then averaged or on a time series of the spatial average of rainfall, or something else?
Answer from the authors:
The decorrelation time is derived by computing the correlation coefficient of precipitation fields for increasing time lags, for example $\rho(R(t), R(T-1))$, $\rho(R(t), R(T-2))$, ..., $\rho(R(t), R(T-N))$, where R is the rainfall field and N is a sufficiently large time lag to allow full decorrelation of the precipitation fields. We added more explanations in the manuscript and a reference.

Question from reviewer:
- l. 210: may be add a percentage?
Answer from the authors:
Nice suggestion. The number is 33 %, which means that over the Swiss domain there is precipitation somewhere one third of the time.

Question from reviewer:
- l. 223-229: I have to admit that I have some trouble in understanding what is plotted. May be describing more precisely the plots and what each of them shows / brings would be helpful.
Answer from the authors:
We rephrased the whole paragraph to clarify that the surfaces of sections are practically a cross-section through the attractor, which is obtained by selecting all the points within a certain interval from the median of the chosen variable.

Question from reviewer:
- Section 3.3: why WAR was used to "explain" beta_2-beta_1 and not another variable?
Answer from the authors:
The most useful descriptor was not the WAR alone, but rather the WAR normalized by the MM (marginal mean conditioned to only precipitation areas). This parameter is high for widespread stratiform precipitation (high WAR, low MM) and low for convective precipitation (low WAR, high MM). This also means that convective precipitation fields exhibit a weaker scaling behaviour than large-scale stratiform precipitation systems. It is then not surprising that nowcasting of thunderstorms is done by cell-tracking techniques rather than scaling approaches. We expanded the text accordingly.

Question from reviewer:
- Figure 8: Is it s_1 (eq. 2) that is plotted? How are the analogues found and how are averages over all the analogues computed? It is somehow related to l.267-268. I do not really understand why the fact that there is less analogues decreases predictability? Once they are found, we can expect they remain closer…
Answer from the authors:
Correct: we are using Eq. (2) to compute the spread (we specified this in the revised version of the paper).
Instead of using advanced analogue selection techniques (e.g. Foresti et al. 2015, Atencia et al. 2015), for this experiment we retrieved the analogues by dividing each dimension of the attractor into 10 intervals (from minimum to maximum value). More precisely, a 1D attractor has 10 intervals, a 2D attractor 100 squares, a 3D attractor 1'000 cubes and a 4D attractor 10'000 hypercubes. The points that fall within the same hypercube are considered analogues. Cubes with less than 20 analogues were discarded. The revised paper was extended with this information.
We unfortunately do not understand the last question as we did not discuss this in text (less analogues = less predictability). Rather, we concluded that less phase space variables = less predictability in the 6h-20d range.

Question from reviewer:
- l.287-290: have you considered the differences in spatial and temporal resolution of the data (1 km x 5 min vs. 64 km x 15 min) as a potential cause for explaining the differences in predictability?
Answer from the authors:
Interesting question. We do not expect the temporal resolution of the data to have an impact on the predictability analyses (apart from reducing the granularity of the temporal dimension). The different spatial resolution might have an impact on the derived global statistics and their predictability, although we did not perform an upscaled analysis on the Swiss attractor (with a pixel resolution of 64x64 km, this would give only an 8x8 radar image, which is too small to compute e.g. Fourier spectra).

We mainly attribute the longer predictability estimates on the US attractor to the fact that it is possible to observe the full precipitation extent and lifecycle of extra-tropical cyclones, while in Switzerland we only see a portion of it, which more rapidly enters and leaves the radar domain. The manuscript was extended accordingly.

Question from reviewer:
- l. 315-317: I guess it changes with regards to comparison with the Swiss case which is then focusing on much smaller scales (1 km vs 64 km).
Answer from the authors:
We did not attempt to upscale the Swiss radar domain to the resolution of 64x64 km$^2$ pixels as this would have given an 8x8 radar image, too small to provide a meaningful comparison to the US dataset. We added this comment in the revised manuscript.

Question from reviewer:
- l. 368-369: could you simply provide the reader with few details on how the events were selected.
Answer from the authors:
The selection included well-organized precipitation systems moving across the continental US from West to East. We extended the revised manuscript accordingly.

Question from reviewer:
- section 4.4: I have trouble to see the pseudo rectangular trajectories that are mentioned in the comments... I do not really understand what the authors are obtaining from Fig. 12. Could you please clarify?
Answer from the authors:
We renamed pseudo-regular patterns into quasi-regular trajectories. This behaviour is explained in Sect 4.2: the 2$^{nd}$ and 3$^{rd}$ principal components describe the general location of the precipitation system within the domain, as previously observed by Foresti et al. (2015). We adapted the revised manuscript accordingly.

Question from reviewer:
- Fig. 15: please clarify how the time series of PC are obtained for each event (same events as in Fig. 12?)
Answer from the authors:
Fig. 15 is from the Swiss attractor and Fig. 12 from the US attractor. So, they are not the same events. Instead of computing a single autocorrelation function for the whole time series (6 years of principal component values), we computed it for each precipitation event (separated by a sufficiently long period without precipitation). This step was needed because the (long) periods without precipitation were artificially increasing the temporal autocorrelation. We extended the revised manuscript accordingly.

Question from reviewer:
- l. 447: "one" should be "once" I guess
Answer from the authors:
Thanks for noticing the typo. Corrected.

Question from reviewer:
- Appendix E is missing (there is only the title)
Answer from the authors:
Thanks. We added one sentence to refer to the figure.

**Reviewer 2 (Citation: https://doi.org/10.5194/npg-2023-24-RC2)**

Summary
The authors analyze time series of rain radar patterns and perform an analysis in terms of low-dimensional deterministic structures. This is a very interesting endeavor, in particular, as this study compares results obtained for two different regions of the globe.

While I consider the results to be inconclusive, I appreciate the effort, because it provides insight into the strengths and weaknesses of different approaches to data compression, in particular by the careful comparison of representations by different interpretable statistical quantifiers and representation in a time delay embedding space.

I also appreciate highly that the authors explicitly do not claim to have detected a low dimensional attractor. Instead, the study shows the potential and the limitations of low dimensional representations.

Answer from the authors:
We are glad to hear that you found the paper interesting and thank you for the comments provided below, which helped improving the manuscript.

Question from reviewer:
I do have some remarks which might help to sharpen the message and to remove some weaknesses of the paper:

1. The terminology: The authors recurrently call the different variables by which they represent the actual state of the radar image as "phase space dimensions". Although I understand what they mean by this, my mathematical education feels a bit offended by that, since the "dimension of the phase space" is a fixed number, namely the number of coordinate axes I need to represent a state vector in phase space uniquely. Instead, the authors use this term for the different coordinates used to represent (a subset of) the phase space. I therefore suggest to rephrase, e.g., the heading of section 3.1, 4.2, and related phrases in the text.
Answer from the authors:
Thanks for pointing out this terminology issue. We replaced all the terms "phase space dimensions" with "phase space variables" in the manuscript. This way we avoid any confusion with the dimensionality (dimension) of the phase space nor the fractal dimension of the attractor.

2. error growth: For deterministic chaotic systems, where an exponential growth of errors is expected, the standard way to represent error growth over time is to plot the logarithm of the error versus a linear time axis. The exponential part of the error growth then results in a linear increase in such a plot. The authors certainly have good reasons to use a logarithmic compression of the time axis (the time covered extends across several orders of magnitude) and to chose a linear error-axis (the error varies only by a factor of 10), but this way of plotting is unusual, and the "classical one" might be useful as an inset, in particular to see or not to see some initial exponential growth due to classical chaos. Overall, I find the improved predictability for lead time up to 10 days quite remarkable (Fig.8). Does the range of power-law error growth for the Swiss data match the power law range (in time) for the US-data?

Answer from the authors:

Thanks for the comment. As you mention, the logarithmic compression of the time axis was done to highlight the longer range of predictability. As suggested, we regenerated one Figure (Fig. 9) using a standard linear time vs log error scaling to check whether there is a linear increase in the first few hours, which would represent an exponential error growth (in a linear time vs linear error plot). The results are not showing a clear linearity at the start, except for a limited number of phase space variable and initial conditions. Therefore, we cannot claim that there is an exponential error growth. The new plot was added to the Appendix of the revised paper, where we also included other scaling versions of Fig 9. To account for reviewer 1 comments, we also moved the current version of Fig. 9 to the Appendix and replaced it with a version using the same scaling as Fig. 8 (log time vs linear error). By comparing Fig. 8 and Fig. 9 we notice that the regions of fast error growth in range 1-6 h correspond quite well, although the Swiss attractor shows a less clear breaking point at around 6 h compared to the US.

Question from reviewer:
3. Fig. 12 is quite interesting. I suggest to mark the beginning of the trajectories with some symbol.

Answer from the authors:

Thanks for the suggestion. Unfortunately, this figure is not anymore reproducible as the US dataset is not anymore accessible to the authors.

Question from reviewer:
4. Discussion of deterministic features of a spatially extended system: A typical feature of rain radar images are moving fronts. A moving front has a rather low dimensional dynamics: on short times its spatial orientation and the propagation speed vector are sufficient. Are the variables used here suitable to represent such a low dimensional object also as low dimension in the reconstructed phase space? It is my impression that our common tools of data analysis are not well designed to capture structures which are

just moving with little change of their other properties, and this seems to me to be particularly true for EOFs.

Answer from the authors:

PCA (EOFs) is suitable to describe the general location of a moving front, in particular the principal components 2 and 3, as explained in Sect. 4.2. In the revised manuscript version, we reinforced this statement in Sect. 4.4 to explain that the faster the precipitation system moves, e.g. from West to East, the faster is the trajectory along the $2^{nd}$ principal component.

In the domain-scale precipitation attractor, we computed the anisotropy of the precipitation field, but did not take into account its orientation. We find your suggestion of adding orientation and translation speed as phase space variables very promising for future developments. We mentioned it as possible extension in the conclusion of the revised manuscript.